# Molecular and cellular determinants of motor asymmetry in zebrafish

Eric J. Horstick[1,2✉], Yared Bayleyen[1] & Harold A. Burgess [1✉]

Asymmetries in motor behavior, such as human hand preference, are observed throughout bilateria. However, neural substrates and developmental signaling pathways that impose underlying functional lateralization on a broadly symmetric nervous system are unknown. Here we report that in the absence of over-riding visual information, zebrafish larvae show intrinsic lateralized motor behavior that is mediated by a cluster of 60 posterior tuberculum (PT) neurons in the forebrain. PT neurons impose motor bias via a projection through the habenular commissure. Acquisition of left/right identity is disrupted by heterozygous mutations in *mosaic eyes* and *mindbomb*, genes that regulate Notch signaling. These results define the neuronal substrate for motor asymmetry in a vertebrate and support the idea that haploinsufficiency for genes in a core developmental pathway destabilizes left/right identity.

[1] Division of Developmental Biology, Eunice Kennedy Shriver National Institute of Child Health and Human Development, Bethesda, MD 20892, USA.
[2] Present address: Department of Biology, West Virginia University, Morgantown, WV, USA. ✉email: eric.horstick@mail.wvu.edu; burgessha@mail.nih.gov

In many bilaterian organisms, the nervous system shows striking structural and functional lateralization, and many species show prominent asymmetric motor behaviors such as hand or paw preference[1]. However, attempts to link neuroanatomic and motor asymmetries have yielded contradictory and inconclusive results in humans[2–4]. In other species, despite correlations between asymmetric properties of the nervous system and lateralized behavior, causal relationships have not been established[5–7]. In the absence of a compelling neuronal basis, it has been difficult to resolve molecular determinants that drive the development of lateralized behavior.

Research to date has produced limited insight into the molecular mechanisms that establish motor asymmetries. Consistent with early theories that outlined a role for single genes of large effect, twin studies on the heritability of handedness in humans revealed an important genetic component[8,9]. However, genome-wide genetic studies failed to uncover loci with large contributions[10,11]. Behavioral hand preference is established as early as 10 weeks of gestation in humans, and left/right asymmetries in gene expression within the spinal cord have been identified at this stage, suggesting a role for genes that pattern the nervous system[12,13]. Abnormalities in neuroanatomical asymmetry and handedness are associated with schizophrenia and other neurodevelopmental disorders; however, without knowledge of the underlying neuroanatomical substrates or developmental pathways, these findings are difficult to interpret[14,15].

In the absence of a clear structural basis for a vertebrate motor asymmetry, many studies have instead focused on molecular genetic pathways that govern the development of brain asymmetries. In particular, work in zebrafish has outlined molecular genetic pathways that govern asymmetric morphogenesis and gene expression within the dorsal diencephalon. In zebrafish, as in many other vertebrates, the habenula shows a pronounced hemispheric asymmetry with well-characterized differences in the size, composition, and connectivity of subnuclei on the left and right sides[16–20]. Moreover, functional differences between hemispheres are also apparent, with olfactory cues preferentially activating the right habenula and visual stimuli driving responses in the left habenula[21–24]. Behaviorally, the habenula in fish has been implicated in social conflict, anxiety, and avoidance learning[25–28], and the left habenula specifically in attenuating fear and driving light-preference behavior[29]. However, to date, assays reporting individual lateralized motor behavior in zebrafish have proven too difficult to reproduce, precluding efforts to resolve the underlying asymmetries in brain structure[30,31].

Here, we report that individual larval zebrafish show a consistent motor asymmetry across multiple behavioral assays when tested in the absence of visual stimuli. Motor identity is maintained by a cluster of 60 neurons in the rostral lobe of the posterior tuberculum (PT) that project to the habenula nuclei, the ablation of which also degrades motor bias. Finally, we demonstrate that lateralized behavior is disrupted by haploinsufficient mutations in genes that regulate Notch signaling.

## Results

**Zebrafish show individual lateralized circling behavior.** After loss of illumination, 6 days post fertilization (dpf), larval zebrafish initiate a circular swimming behavior in which they repeatedly perform same-direction turn maneuvers in a restricted spatial area (Fig. 1a)[32]. Same-direction turn movements were sustained in individual larvae for 2 min (Fig. 1b); however, across the population, there was no net tendency for larvae to preferentially circle to the left or right (mean net turn angle (NTA) = $-60.8 \pm 84.2$, one-sample $t$ test against 0, ±standard error of the mean, $p = 0.24$, Supplementary Fig. 1a). However, we asked whether individual larvae show lateralized behavior—preferentially swimming in the left or right direction—or randomly select a circling direction on each event. We therefore probed individuals with a series of four light-off trials, each separated by several minutes of illumination. Larvae that circled in a rightward direction on the first trial, showed a significant tendency to circle rightward on subsequent trials, and similarly, left-circling larvae continued to show a leftward bias (repeated measures ANOVA, main effect of trial 1 direction $F_{(1,65)} = 22.20$, $p < 0.001$, $\eta^2_p = 0.25$, Fig. 1c). Lateralized behavior was not observed when larvae were tested under constant illumination, as groups initially classified as left/right based on the first trial did not show differences in mean direction on subsequent trials (repeated measures ANOVA $F_{(1,61)} = 1.08$, $p = 0.30$, Fig. 1d). Directional bias on dark trials manifest as a 69.6% probability for larvae to circle in the same direction as on the first trial (match index, one-sample permutation test, $p < 0.0004$ compared with 50% probability, Fig. 1e). Circular swimming is primarily driven by routine-turn maneuvers[32]. We therefore used high-speed video recordings and kinematic analysis to directly measure routine-turn direction across a series of four light-off trials (Fig. 1f). Again, we noted that turn direction on trials 2–4 was significantly correlated with turn direction on the first trial (Supplementary Fig. 1b, d). Next, we calculated the percentage of routine turns made in a rightward direction on each trial, then used the mean of the four trials to represent each larva's direction preference. In the dark, the distribution of direction preferences strongly deviated from the expected distribution: 41% of larvae (37/89) produced fewer than 24% or greater than 76% rightward turns, whereas if turn direction was random on each trial, only 10% of larvae would have shown this level of bias (Monte Carlo simulations, $p < 0.0001$, Fig. 1g). In contrast, under constant illumination, the distribution of routine-turn direction bias was similar to the expected distribution (Monte Carlo simulations, $p = 0.119$, Fig. 1h). These results reveal that zebrafish larvae raised in the same environment stochastically acquire a left/right directional bias in motor behavior that is manifest when tested under dark conditions.

A form of unstable lateralized eye-use behavior has been reported in zebrafish, with individuals switching eye preference over several minutes[33]. In contrast, we found that turn bias during dark-induced circling behavior was sustained for at least 45 min (Supplementary Fig. 1c). Next, we tested individuals at 6 dpf to establish their left/right motor identity, then returned them to an incubator overnight. The next day, we recorded responses to a second set of four light-off stimuli. Turn direction bias for individual larvae was significantly correlated between days (Spearman rho = 0.58, $p = 0.0009$, Fig. 1i, j, Supplementary Fig. 1e), and also sustained in larvae that were raised for 4 days between tests (rho = 0.39, $p = 0.0003$, Fig. 1j, Supplementary Fig. 1g). Decomposing turn bias into its magnitude and direction components revealed that the overall left/right direction preference was maintained (d6–d7, Mann–Whitney $U$, $p = 0.012$; d6–d10, $p = 0.00039$, Fig. 1j) although the strength of the preference was not correlated across days (d6–d7, rho = $-0.02$, $p = 0.9$; d6–d10, rho = 0.054, $p = 0.7$, Supplementary Fig. 1f, h). Thus, left/right motor identity is sustained in individual larvae for several days. In birds, stable perceptual asymmetries are conferred by visual experience during embryonic development[34]. In contrast, dark-reared zebrafish larvae showed normal lateralized behavior, excluding an instructive role for visual experience in the acquisition of motor identity (Supplementary Fig. 1i). Together, these data provide compelling evidence that zebrafish larvae show a robust motor asymmetry, manifest as persistent individual differences in the direction of circling behavior after loss of illumination. Behavioral asymmetry in zebrafish is of moderate

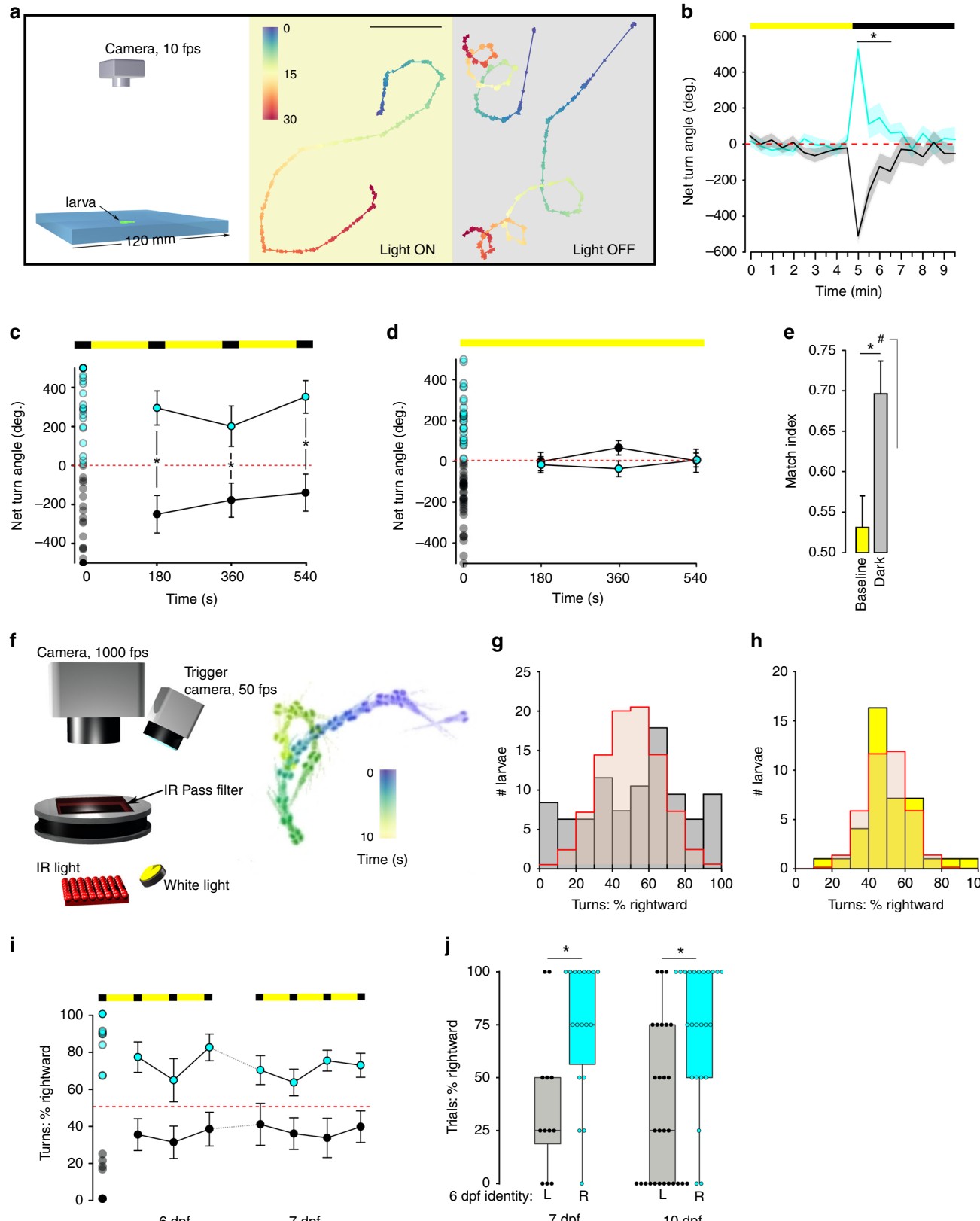

strength: larvae initiate movement in their preferred direction on around 70% of trials. Similar numbers of larvae are left- and right-biased, with no apparent population-level asymmetry.

Next, we asked whether motor bias is also present in responses to other stimuli. Zebrafish show positive phototaxis, and select and swim toward one of two simultaneously illuminated regions[35]. We first classified larvae as left- or right-biased during dark-induced circular swimming, then used a closed-loop system to present freely swimming larvae with two identical target spots on the left and right after the loss of full-field illumination (Fig. 2a, b). Target choice was positively correlated with circular swimming direction (Fig. 2c). We also

**Fig. 1 Individual larvae show persistent motor asymmetry during dark-induced circling. a** Swim trajectories during baseline illumination and after loss of light. Color scale: time (seconds). Arrowheads, orientation. Scale bar 20 mm. **b** NTA over 30-s intervals for larvae before/after loss of illumination. Individuals were classified as right- (cyan, $N = 25$) or left-biased (gray, $N = 34$) based on the first 30-s interval. Asterisk $p < 0.05$, $d = 3.1$, 0.71, 0.66, and 0.61 (first four time points in the dark), $t$ test between groups. **c** NTA for 30-s light-off trials (black bars). Open circles at time 0 show individual larvae, and were used to classify as right- (cyan, $N = 34$) or left-biased (black, $N = 34$). Subsequent points show mean and standard deviation for left/right groups. Asterisk $p < 0.05$, $t$ test between groups. **d** As for (**c**) during constant illumination (right, $N = 29$; left, $N = 35$). **e** Match Index during baseline illumination (yellow, $N = 64$) and after loss of illumination (gray, $N = 68$). Asterisk $p < 0.05$, $r = 0.29$, Mann–Whitney $U$ test, and #$p < 0.05$, $r = 0.76$, one-sample permutation test against 0.5. **f** Visually isolated chamber. Right: time-lapse montage over 10 s following loss of illumination. Color: time (seconds). **g**, **h** Percentage of turns executed rightward (mean of four 10-s trials) after loss of illumination (**g**, gray, $N = 89$) or during constant illumination (**h**, yellow, $N = 39$). Red line: expected distribution, Monte Carlo simulation of unbiased larvae. **i** Rightward-turn preference over 24 h. Larvae with <33% of rightward turns at 6 dpf were classified as left-biased (black, $N = 12$), and those >66% as right-biased (cyan, $N = 14$). At 7 dpf, % rightward-turn use in L/R-classified groups. Repeated measures ANOVA for 7-dpf trials, the effect of 6-dpf first-trial direction $F_{1,24} = 15.4$, $p < 0.001$, $\eta^2_p = 0.39$. Asterisk $p < 0.05$ between groups. **j** Percentage trials with net rightward bias for larvae tested at 7 or 10 dpf ($N = 30$, 52), after left/right classification at 6 dpf. Asterisk $p < 0.05$, $r = 0.61$ and 0.60 (7 and 10 dpf), Mann–Whitney $U$ test. Error bars: standard error of the mean. Box plots show median and quartiles with whiskers indicating 10–90%. Source data are provided as a Source Data file.

tested whether lateralized motor behavior occurred during responses to a nonvisual stimulus. In zebrafish, intense auditory stimuli elicit escape responses that are initiated with a rapid bend to either the left or right, raising the possibility that startle direction might show a motor asymmetry. We again pre-classified larvae using dark-induced circular swimming, then measured startle direction. Under constant illumination, escape response direction showed little or no correlation with circular swim direction (Fig. 2d, yellow background). However, auditory stimuli that were presented in the dark evoked startle responses whose direction showed a significant match to the direction of circular swimming (Fig. 2d, gray background). Thus, in the absence of visual cues, motor bias was apparent in different assays, consistent with individual larvae having an intrinsic left/right motor identity.

In each assay, motor asymmetry was primarily manifest under dark conditions, suggesting that visual cues override intrinsic bias. We tested this idea using *atoh7* mutants that completely lack projections from the retina to the brain and in enucleated larvae. Even without retinal signaling, zebrafish respond to changes in whole-field illumination via deep-brain photoreceptors[32,36]. Thus, on trials 2–4 of the dark-induced circling assay, *atoh7* mutants and enucleated larvae demonstrated a significant tendency to swim in the same direction as in trial 1 as for sibling controls (Fig. 2e, gray bars, Supplementary Fig. 1j). However, unlike wild-type siblings, *atoh7* mutants also showed a weak but significant motor asymmetry under illuminated conditions (one-sample permutation test, $p = 0.003$, Fig. 2e, yellow bars). Next, we used acute unilateral enucleation to control the source of visual information. After acute unilateral enucleation, circling was performed in the direction contralateral to the intact eye on all four trials (Fig. 2f). Together, these results support the idea that visual information, when present, predominates over intrinsic motor asymmetry.

**PT neurons maintain motor identity.** We next set out to identify the underlying neuronal substrates for left/right bias. Whereas the parapineal and habenula show consistent population-wise anatomical asymmetries, no brain regions are known in zebrafish to have stochastic hemispheric differences in size or function that would be consistent with individual left/right motor bias[19,37]. We therefore performed a circuit-breaking screen to identify neuronal substrates, crossing Gal4 lines that label distinct brain structures to a UAS:epNTR-TagRFP reporter for cell-specific ablation using nitroreductase (Fig. 3a, Supplementary Table 1)[38,39]. After lesioning the labeled population of neurons in each line, we then tested whether individual motor bias remained. Larvae showed a decrease in motor bias after ablation of neurons in two Gal4 lines:

*y279* and *y375* (Fig. 3b). Reduced motor bias was specific, as ablation did not affect the total amount of turning behavior or baseline locomotion (Supplementary Fig. 2a). Finally, the chemogenetic ablation protocol itself did not disrupt motor bias as ablation of neurons in orthopedia driver line *otpb.A:Gal4* had no effect (Fig. 3b). Thus, the *y279* and *y375* Gal4 lines label neurons that are critical for individual motor bias.

*y279* and *y375* each express Gal4 in neurons distributed in multiple brain regions (Fig. 3c, d). We reasoned that clusters labeled by both lines would be strong candidates for driving lateralized behavior. Because the two lines express the same Gal4 reporter, we could not directly identify co-labeled neurons, but instead compared co-registered whole-brain images of each line[40]. Salient virtual co-expression was present in a small bilateral cluster of neurons in the rostral domain of the PT, the fish derivative of the basal plate of prosomere 3 (Fig. 3e, see Supplementary Fig. 2b–e for comparative neuroanatomical discussion). Cell counts established that each PT hemisphere comprised $28.3 \pm 7.2$ $y279^+$ cells (mean ± standard deviation, $N = 28$ larvae), but did not reveal differences between hemispheres in left- and right-biased larvae (Supplementary Fig. 2f). We did not observe co-expression of either an excitatory (*vglut2a*) or inhibitory neurotransmitter (*gad1b*) transgenic marker in rostral PT neurons (Fig. 3f). We assessed whether the PT clusters were required for left/right bias by using focal laser ablation of *y279* neurons. As a control, we also ablated the caudal hypothalamus (Hc), which is strongly labeled in *y279*. Ablation of the Hc had no effect, whereas bilateral ablation of the rostral PT cluster strongly reduced motor bias, consistent with a loss of left/right identity (Fig. 3g, Supplementary Fig. 3a). Conversely, after unilateral PT ablations, 76% of larvae (38/50) circled in the direction ipsilateral to the intact PT during dark trials, with no effect during baseline illumination (Fig. 3h, Supplementary Fig. 3b, c). Lateralized responses after unilateral PT ablation were also observed for startle behavior in the absence of illumination (Fig. 3i). The PT is a ventral region of the diencephalon that includes identified classes of dopaminergic (DA) neurons[41]. However, the cluster of PT neurons labeled in *y279* and *y375* is situated rostral and dorsal to PT DA neurons (Supplementary Fig. 2d, *th*+). Moreover, *otpa* mutants, which lack key classes of DA neurons, and *otpb.A:Gal4* ablations, retained robust left/right bias (Fig. 3b; Supplementary Fig. 3d). It is therefore unlikely that PT DA function contributes to motor asymmetry. These results confirm that a cluster of neurons in a rostral lobe of the PT impose left/right bias on motor behavior in zebrafish.

**PT neurons show sustained activity after light extinction.** We next examined whether PT neurons are active after loss of

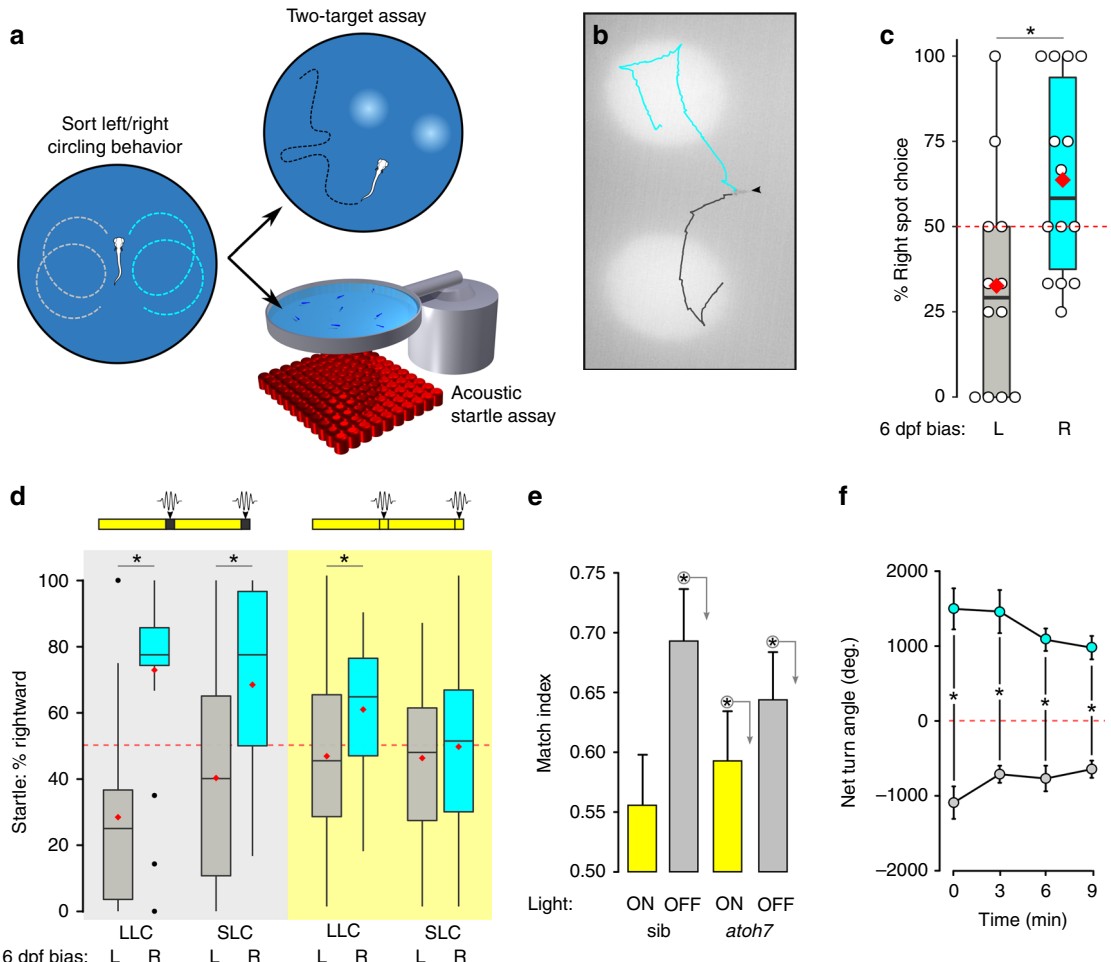

**Fig. 2 Motor asymmetry is correlated across multiple behavioral tasks. a** Experimental paradigm: larvae were classified as left-/right-biased at 6 dpf based on circling response after loss of illumination (over 4 trials). At 7 dpf, the same larvae were tested in either a two-target phototaxis assay or acoustic startle assay. **b** Representative path trajectories for right- (cyan) and left- (gray) classified larvae (arrowhead) presented with two symmetric light spots following loss of illumination. **c** Percent of trials on which larvae turned toward the right spot, for larvae classified at 6 dpf as left-biased (gray, $N = 12$) and right-biased (cyan bar, $N = 13$). Each larva performed four trials, with trials excluded if the larva was adjacent to the arena edge when light spots were presented. Asterisk $p < 0.05$, $r = 0.51$, Mann–Whitney $U$ test. **d** Percentage of startle responses made in a rightward direction for larvae preclassified as left- (gray) or right- (cyan) biased. Larvae were tested either in the dark (gray background) or light (yellow background) conditions. As acoustic stimuli elicit either short- or long-latency C starts (SLC, LLC) that are mediated by different circuits, response types were analyzed separately. Red diamond indicates mean. Dark LLC responses: left $N = 19$, right $N = 20$. Dark SLC responses: left $N = 12$, right $N = 14$; light LLC responses: left $N = 27$, right $N = 23$; light SLC responses: left $N = 18$, right $N = 28$. Asterisk $p < 0.05$, $r = 0.71$, 0.51, and 0.35, respectively, Mann–Whitney $U$ test. **e** Match index for *atoh7* mutants and siblings during baseline illumination (sib, $N = 45$; mutant, $N = 57$; yellow bars) and dark conditions (sib, $N = 51$; mutant, $N = 58$; gray bars). Asterisk $p < 0.05$, $r = 0.74$, 0.20, and 0.41, respectively, one-sample permutation test to 0.5. **f** NTA for each of four trials after unilateral enucleation of the left (gray, $N = 4$) or right (cyan, $N = 6$) eye. Dotted red line: random output. Repeated measures ANOVA, the effect of side-lesioned, $F_{1,7} = 115.8$, $p < 0.001$, $\eta^2_p = 0.94$. Asterisk $p < 0.05$ between groups. Error bars represent standard error of the mean. Box plots show median and quartiles with whiskers indicating 10–90%. Source data are provided as a Source Data file.

illumination by imaging changes in GCaMP6s fluorescence in *y279-Gal4, UAS:GCaMP6s* larvae. Because we previously found that photic stimulation acutely terminates localized circling behavior[32], we used an infrared laser for two-photon excitation of GCaMP6s during these recordings. We attempted to simultaneously record tail movements, but found that persistent motor responses to the loss of illumination were not apparent in immobilized larvae. Nevertheless, we reasoned that even if the motor behavior was not preserved, PT neurons might still respond to loss of illumination. Indeed, 15.3% of PT neurons (46/299 from 10 larvae: 4 left, 6 right motor-biased) responded to the light-off stimulus (Fig. 4a, b), with activity peaking shortly after loss of illumination. Consistent with the time frame of circling behavior (Fig. 1b, see above), more than half (27/46) of the OFF-responsive

neurons returned to half-maximal activity within 30 s of the light OFF stimulus, with a mean time to half-max activity of 22.4 ± 14 s (mean/std. dev.). Interestingly, activity remained elevated above baseline thereafter, even during prolonged dark periods, possibly accounting for the lower level of motor bias seen for several minutes after loss of illumination (Fig. 4c, d). An additional 4.7% of PT neurons (14/299) were active upon the restoration of illumination. However, no significant correlations were observed between the left/right identity of individual larvae and peak GCaMP activation, position of light OFF-responsive neurons within the PT, latency to firing initiation, or time to peak activity (Fig. 4e, f, Supplementary Fig. 2g, h). Since PT neurons were acutely responsive to changes in illumination, we asked whether PT neurons might receive photic information directly from the

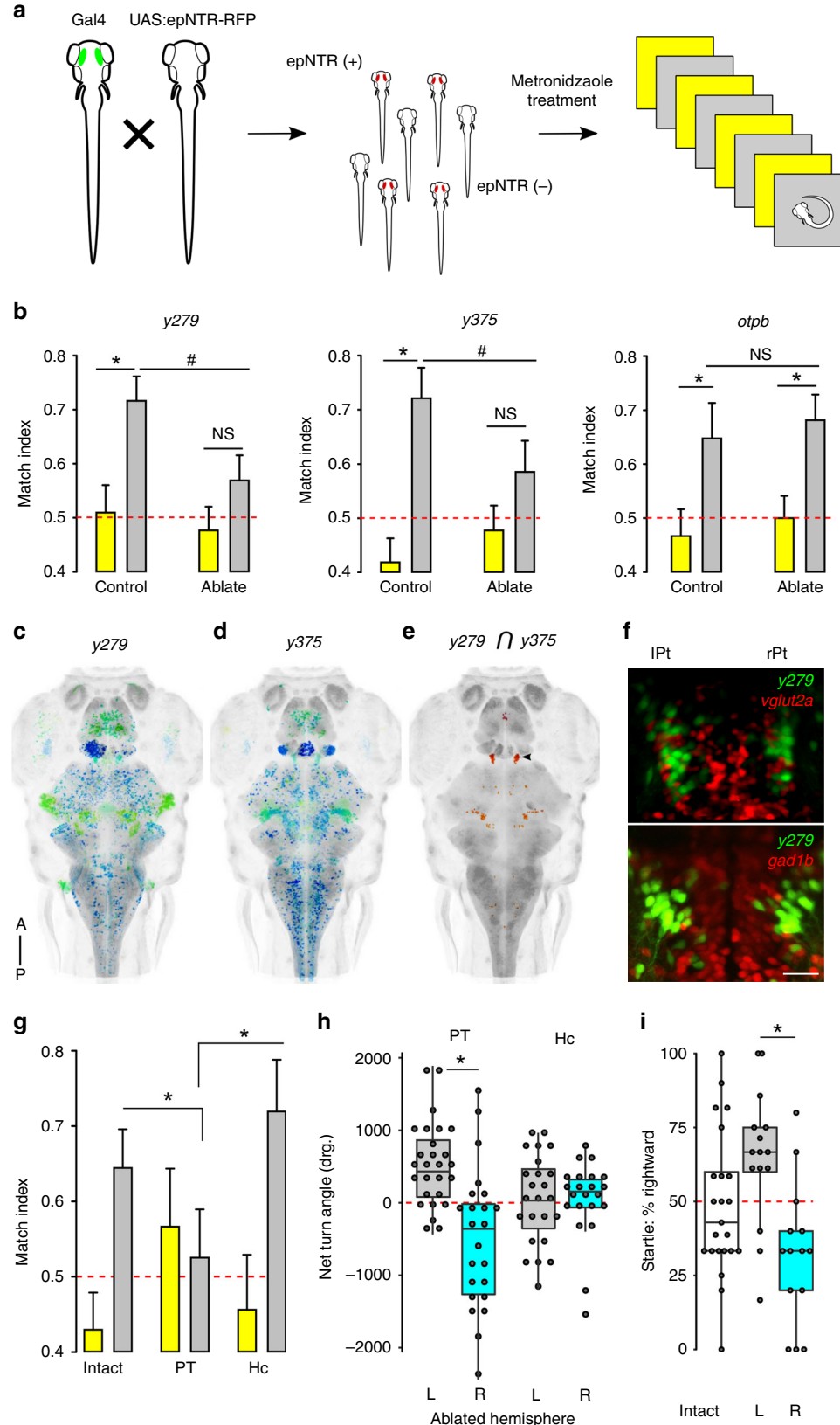

eye. We visualized retinal ganglion cell (RGC) termination zones (arborization fields, AF) in *atoh7:GFP* larvae, and found that rostral PT neurons were proximal to region AF3 (Supplementary Fig. 2i)[42]. RGC axons did not directly ramify within the rostral PT; however, neurites from RGCs and the PT were closely apposed

(Supplementary Fig. 2j). These results establish that a subset of PT neurons fire in response to changes in illumination, remain active on a timescale consistent with the duration of motor asymmetry during dark-induced circling behavior, and potentially receive direct photic input from the retina.

**Fig. 3 Neurons in the posterior tuberculum maintain left/right identity. a** Chemogenetic ablation screen: transgenic lines with restricted Gal4 patterns were crossed to a UAS:epNTR-RFP reporter. Both epNTR-RFP$^+$ and non-fluorescent siblings (as controls) were treated with metronidazole before testing for motor asymmetry under light and dark conditions. **b** Match index for drug-treated controls (y279 $N = 53$; y375 $N = 43$; otpbA $N = 35$) and following genetic ablation (y279 $N = 57$; y375 $N = 37$; otpbA $N = 46$) during paired baseline (yellow) and dark (gray) responses. Asterisk $p < 0.05$, $r = 0.31, 0.49, 0.3$, and $0.34$, respectively (left to right); $^{\#}p < 0.05$, $r = 0.25$, Mann–Whitney U test. **c, d** Whole-brain dorsal Zebrafish Brain Browser (ZBB) projections for y279 (**c**) and y375 (**d**). Color is depth scale. **e** Computed intersect y279 and y375 expression patterns. Arrowhead indicates a cluster in the rostral PT. **f** Dorsal confocal projection through the rostral PT in (top) y279-Gal4, UAS:Kaede (green) crossed to vglut2a:dsRed (red) or (bottom) y279-Gal4, UAS:Kaede (green) crossed to gad1b:dsRed (red). Scale bar 20 μm. **g** Match index in unablated controls ($N = 45$) and after bilateral laser ablation of the PT ($N = 17$) and Hc ($N = 19$) during baseline (yellow) and on dark trials (gray). Asterisk $p < 0.05$, $r = 0.44$ and $0.59$, respectively, Mann–Whitney U test. **h** Net turn angle (mean on trials 1–4) for the left PT hemisphere (gray, $N = 27$) and the right PT hemisphere (cyan, $N = 23$) ablations. Hc unilateral ablations (right bars) (left hemisphere gray, $N = 24$; right hemisphere cyan, $N = 22$). Asterisk $p < 0.001$, $d = 1.3$, t test. **i** Percentage of dark evoked long-latency startle responses initiated in the rightward direction for intact controls ($N = 23$), left ($N = 15$) or right ($N = 14$) hemisphere PT ablation. Asterisk $p < 0.001$, $d = 1.4$, t test. Error bars represent standard error of the mean. Box plots show median and quartiles with whiskers indicating 10–90%. Source data are provided as a Source Data file.

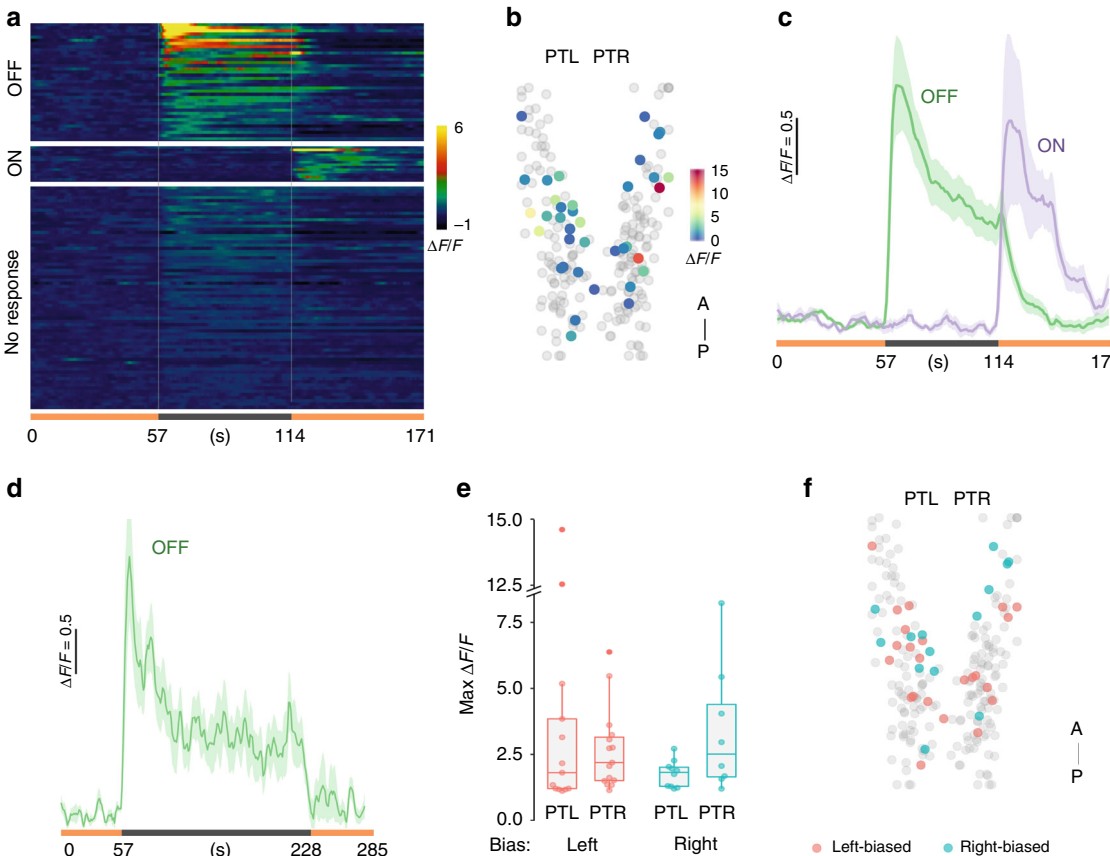

**Fig. 4 Neurons in the posterior tuberculum respond to changes in illumination. a** Raster plot of mean calcium responses (mean of three trials) from GCaMP6s expressing PT neurons. Color scale denotes standardized change in fluorescence intensity ($\Delta F/F$). Illumination conditions as indicated on the X axis (light ON, orange; dark, gray). Only a subset of no-response neurons are included. **b** Location of light OFF-responsive neurons in the rostral PT. Scale bar indicates fluorescence ($\Delta F/F$) change over baseline. **c** Mean and standard error for response of light OFF- (green, $N = 46$) and light ON- (purple, $N = 14$) responsive neurons. **d** Mean and standard error for response of light OFF- (green, $N = 8$) responsive neurons during the 3-min dark period. **e** Peak change in fluorescence for neurons that respond to light OFF in the left (PTL) and right (PTR) PT for larvae classified as left (red) and right (blue) motor-biased. **f** Location of light OFF-responsive neurons within the PT for larvae behaviorally identified as left- (red) or right- (blue) biased. Error bars represent standard error of the mean. Box plots show median and quartiles with whiskers indicating 10–90%. Source data are provided as a Source Data file.

### Projections from PT to habenula are essential for motor bias.

The commissure that runs between the habenula hemispheres is labeled in y279-Gal4, UAS:Kaede larvae (Fig. 5a). As y279 also labels neurons in the habenula—particularly in the left habenula—we initially suspected that this commissure was formed by y279 habenula neurons. However, when performing confocal scans to verify laser ablations, we noticed that fluorescently labeled fibers were absent in bilateral PT-lesioned larvae, raising the possibility that these commissural fibers originate in the PT (Fig. 5b). To test this, we unilaterally photoconverted Kaede in the PT in y279-Gal4, UAS:Kaede larvae. Photoconverted Kaede labeled neurites that emerged dorsally from each PT hemisphere and projected to the ipsilateral habenula. Within the habenula, these fibers turned and crossed to the contralateral side through the habenular

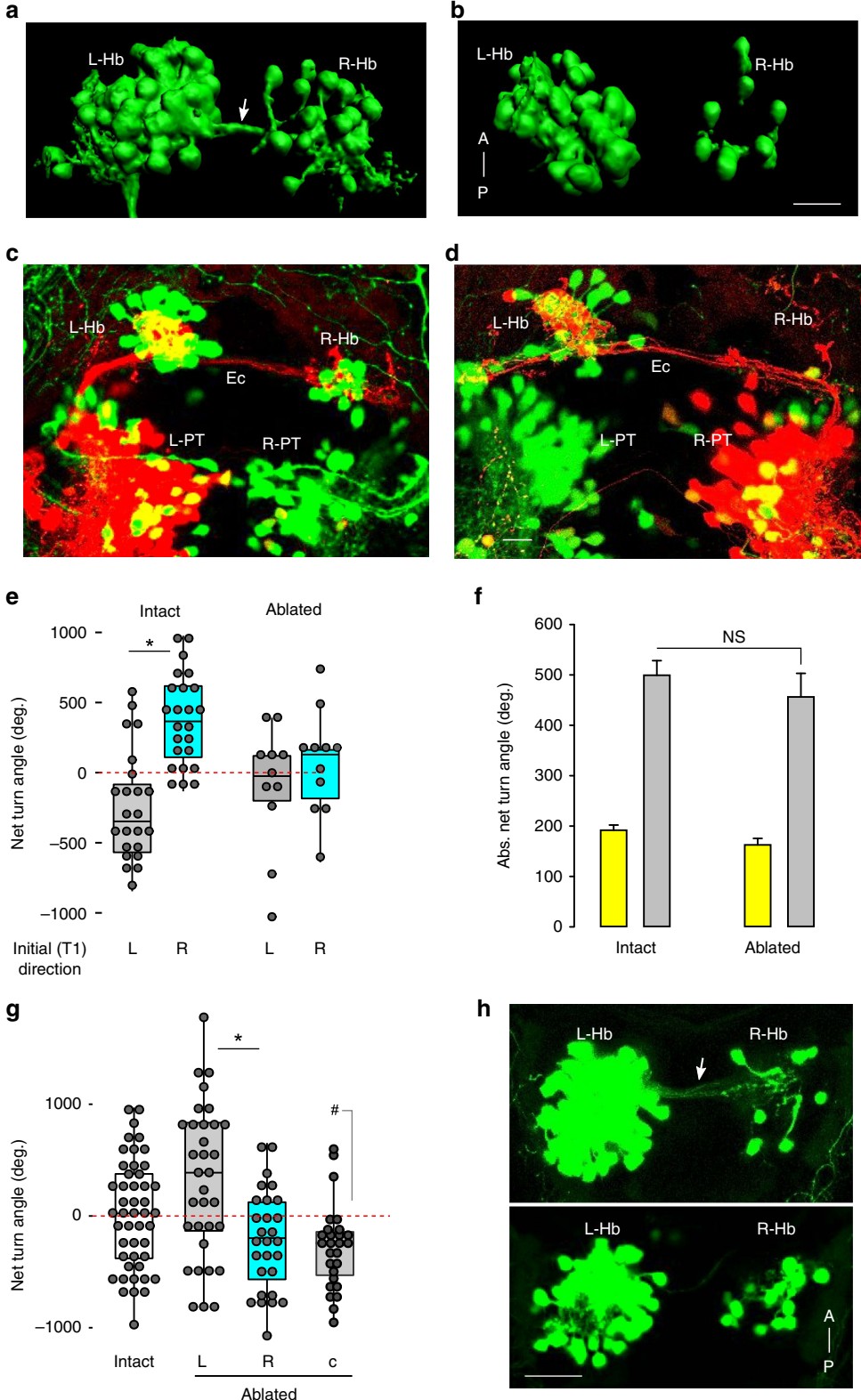

commissure (Fig. 5c, d). PT projections terminated in the neuropil region of both habenula hemispheres with a significant majority terminating in the left habenula, regardless of PT converted (Supplementary Fig. 3e) or motor bias (Supplementary Fig. 3f). Although no salient differences in the PT-habenula projection were observed for left- or right-biased larvae, this observation raised the possibility that the PT controls motor bias

via signaling to the habenula. We tested this by laser ablation of *y279* habenula neurons (Supplementary Fig. 3a). Bilateral ablation of *y279* habenula neurons abolished left/right bias without affecting total turning (Fig. 5e, f). Moreover, after unilateral habenula ablation, motor responses were strongly lateralized during dark-induced circling behavior, without inducing motor bias under baseline illumination (Fig. 5g; Supplementary Fig. 3c).

**Fig. 5 A projection from posterior tuberculum to habenula drives motor asymmetry. a, b** 3D rendering of *y279-Gal4, UAS:Kaede* expression in the habenula at 7 dpf in intact larvae (**a**) and after bilateral ablation of *y279*-expressing PT neurons (**b**). Arrow indicates habenular commissure present in intact larvae that is lost after bilateral PT ablation. Scale bar 20 μm. **c, d** Confocal projections of *y279-Gal4;UAS:Kaede*-expressing larvae following unilateral focal photoconversion of Kaede in the left (**c**) or right (**d**) rostral PT. Photoconverted Kaede in the red channel is saturated to facilitate visualization of PT projections. Scale bar 20 μm. **e** Net turn angle (mean of trials 2–4) for intact control larvae (N = 47) and after bilateral ablation of *y279*-expressing habenula neurons (N = 22). Larvae were classified as left-biased (gray) or right-biased (cyan) based on trial 1 (T1). Asterisk $p < 0.001$, $d = 1.4$, $t$ test. **f** Total amount of turning for larvae in (**e**) during locomotion under baseline illumination (yellow) or dark-induced circling behavior (gray). **g** Net turn angle (mean of trials 1–4) in non-ablated controls (white, N = 47), and following unilateral ablation of the left (gray, N = 33) and right (cyan, N = 28) habenula nuclei, or following laser section of the habenular commissure (c) (N = 24). Asterisk $p < 0.001$, $d = 0.90$, $t$ test. #$p < 0.05$, $d = 0.73$, one-sample $t$ test to 0. **h** Representative confocal scan showing *y279*-labeled habenular commissure (arrow) in a control larva (top). Following ablation commissure is absent (bottom). Scale bar 20 μm. Error bars represent standard error of the mean. Box plots show median and quartiles with whiskers indicating 10–90%. Source data are provided as a Source Data file.

In contrast to PT lesions, total turning behavior did not increase after unilateral habenula ablation (Supplementary Fig. 3g). After unilateral habenula ablation, *y279-Gal4*-labeled fibers within the habenula commissure remained intact, consistent with the conclusion that these fibers originate from the PT (Supplementary Fig. 3h). Intriguingly, selective lesion of the habenular commissure also induced a strong motor asymmetry that was similar to ablation of the right habenula, such that larvae became left-biased only under dark conditions (Fig. 5g, h, Supplementary Fig. 3c, i). Consistent with reports that the habenula is activated by both light ON and OFF cues[21,22,24], calcium imaging revealed a subset (16/101) of *y279-Gal4* habenula neurons that responded during light ON/OFF transitions (Supplementary Fig. 3j). Together, the loss of motor bias after bilateral ablation of either the PT or habenula, induction of left-/right-lateralized behavior after unilateral ablations, and presence of a direct connection between these areas, argues that the PT-habenula pathway imposes left/right bias on motor responses.

**Notch pathway mutations disrupt lateralized behavior.** Many species, including zebrafish, show a stereotyped asymmetric development of specific brain nuclei and placement of viscera. Genes involved in this process have been relatively well characterized[19,43]. However, much less is known about genetic pathways that lead to the stochastic acquisition of left or right motor asymmetry. During the course of our studies, we isolated a background mutation (*y606*) in our wild-type stock that weakened motor bias during circling behavior. In affected clutches, 25% of embryos showed a "curly-up" phenotype that severely effected tail morphology and precluded behavioral testing (Fig. 6a). We speculated that the morphological abnormality represented a homozygous phenotype, with loss of motor asymmetry present in morphologically normal heterozygous larvae (Fig. 6b). RNAseq-based bulk segregant mapping linked the mutation to a 6-Mbp interval on chromosome 9, and revealed a 7.7-fold reduction in the expression of the gene *epb41l5* in this interval in *y606* mutants compared with siblings (Supplementary Fig. 4a, b)[44]. No RNAseq reads mapped to the first two exons, and we were not able to amplify these exons from mutants using genomic PCR (Supplementary Fig. 4c). Consistent with this, genomic PCR revealed a 4.4-kb deletion that excises exons 1 and 2 of *epb41l5*, eliminating the transcription and translation start sites (Supplementary Fig. 4d). *Epb41l5* is the mutated gene in the *mosaic eyes* (*moe^b476^*) mutant that shows a similar curly-up phenotype[45], and *moe^b476^* failed to complement y606, confirming that y606 is a new allele of mosaic eyes.

We incrossed *moe^y606^* heterozygous adults and tested morphologically normal siblings of curly-up mutants in the dark-induced circling assay, then performed post hoc genotyping to distinguish wild-type and heterozygous larvae. Wild-type larvae showed normal motor asymmetry; however, heterozygous

*moe^y606^* larvae showed a significant reduction in left/right bias (Fig. 6c). Other motor parameters and asymmetric development of the epithalamus and viscera were unaffected in heterozygotes (Fig. 6e, f). To corroborate these findings, we used *moe^b476^*, an independent allele that we confirmed eliminates the entire *epb41l5* gene (Supplementary Fig. 5a–c). Heterozygous mutations in *moe^b476^* also disrupted motor bias during circling behavior (Fig. 6d). These results demonstrate that acquisition of left/right identity in zebrafish requires two functional alleles of *epb41l5* in zebrafish.

*Epb41l5* regulates the timing of neurogenesis, interacting with proteins in the *Notch* signaling pathway[46,47]. We therefore reasoned that other heterozygous mutations in Notch signaling proteins might perturb left/right bias and tested larvae with heterozygous mutations in *mind-bomb* (*mib*), an E3 ubiquitin ligase that is essential for notch signaling[48]. Indeed, *mib* heterozygotes showed a severe loss of left/right bias during dark-induced circling (Fig. 6g) without perturbing the total amount of turning (Fig. 6h). Together, these findings reveal that acquisition of left/right motor identity is disrupted by mutations in genes that regulate Notch signaling levels during embryonic development.

## Discussion
Here, we reveal that in the absence of visual information, zebrafish manifest a durable left/right motor bias that is driven by neurons in the PT. Zebrafish do not show a population-level bias, unlike in humans where 90% of the population favor the right hand[49]. Rather, similar to motor asymmetries in many other species, lateralized behavior is manifest as an individual bias to execute movements in a preferred direction on ~70% of trials. Left/right identity is maintained over at least several days, despite handling and changes in the environment. Bilateral clusters of ~30 genetically identified neurons per hemisphere in the rostral lobe of the PT are essential for the expression of this motor asymmetry. This conclusion is supported by loss of lateralized behavior after chemogenetic ablation of rostral PT neurons using two transgenic Gal4 lines that have minimal overlap elsewhere in the brain, and direct laser photoablation of the PT. In addition, unilateral laser ablation of PT neurons was sufficient to create a population of fish whose response direction was heavily biased to the side of the intact PT. Critically, after rostral PT lesions, larvae continued to show persistent turning to one side after loss of illumination, but circling behavior was initiated in a random direction on each trial rather than in a preferred direction. It is therefore unlikely that the rostral PT is part of the sensory pathway that initiates dark-induced circling, but rather suggests that the PT is a locus that maintains left/right identity and imposes this bias on motor responses. Intriguingly, the thalamus, dorsally adjacent to the PT, also relays directional information, sending information about visual dimming cues to the tectum[50].

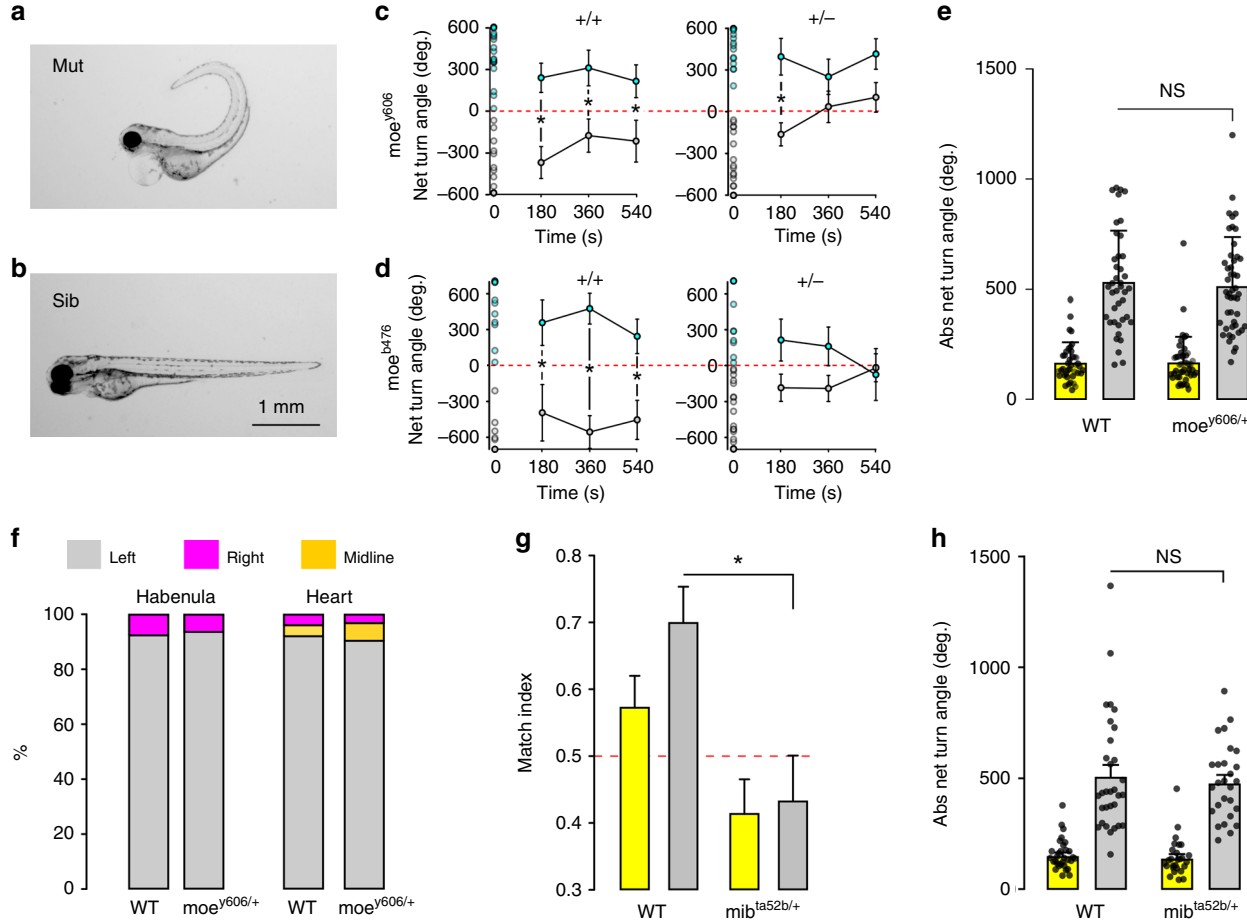

**Fig. 6 Left/right identity is disrupted by mutations in genes that regulate Notch signaling. a**, **b** Curly-up morphology in 2-dpf *y606* mutant (**a**) and sibling larvae (**b**). **c** Net turn angle for *moe^y606* wild-type (left, +/+) and heterozygous (right, +/−) sibling larvae over four 30-s light-off trials. Open circles in the first trial (time 0) represent individual NTA for all larvae tested, and were used to classify larvae as right- (cyan, Het N = 20; WT N = 20) or left-biased (black, Het N = 27; WT N = 20). Subsequent points represent mean for left/right groups on trials 2–4. Repeated measures ANOVA effect of genotype $F_{1,81}$ = 3.8, $p$ = 0.05, $\eta^2_p$ = 0.05. **d** Same analysis as in (**c**) for *moe^b476* allele showing right- (cyan, Het N = 11; WT N = 15) or left-biased (black, Het N = 25; WT N = 9) larvae. Repeated measures ANOVA interaction of genotype/motor-bias $F_{1,56}$ = 5.3, $p<0.05$, $\eta^2_p$ = 0.09. Asterisk $p < 0.05$ between groups in (**c**, **d**). **e** Absolute net turn angle for *moe^y606* heterozygous (N = 47) and wild-type sibling (N = 40) larvae averaged over four trials for baseline illumination (yellow bar) and light-off (gray bar) trials. **f** Habenula and heart placement in wild-type siblings and *moe^y606* heterozygous larvae. Left: percentage of larvae with a larger habenula hemisphere on each side (N = 32 and 25 for wild type, hets). Right: percentage of embryos with the heart positioned on the left, right, or midline (N = 25 and 31 for wild type, hets). **g**, **h** Match index (**g**) and total turning (**h**) for wild-type siblings and *mib^ta52b* heterozygous larvae (N = 31 and 27, respectively), during baseline (yellow) and dark (gray) responses. Asterisk $p < 0.05$, $r = 0.42$, Mann–Whitney U test. Error bars represent standard error of the mean. Source data are provided as a Source Data file.

Finally, neurons in the rostral PT show sustained activity after loss of illumination, consistent with the duration of lateralized circling movements. Together, these findings confirm that the rostral PT drives lateralized behavior in zebrafish, and therefore define a specific neuronal substrate for a lateralized motor behavior in a vertebrate.

Our findings suggest a model in which bilateral PT-habenula units compete to strengthen, but not directly drive ipsilateral dark-induced circling behavior (Fig. 7a). In individual larvae, one hemisphere predominates, providing a greater drive to premotor circuits. How left/right identity is encoded in the PT-habenula unit is unclear: we did not detect differences in the number of neurons or light-off activity between hemispheres that correlated with motor bias. The strong effect of unilateral PT ablations indicates that the intact hemisphere, even if it was not previously dominant, was capable of imposing ipsilateral motor bias, suggesting that dominance is unlikely to reflect a unique quality present in only one hemisphere (Fig. 7b). Moreover, unilateral enucleation results in a robust lateralized response contralateral

to the intact eye, suggesting that visual information overrides intrinsic bias provided by the intact PT (Fig. 7c). However, after unilateral PT ablation, not only was dark-induced motor behavior strongly lateralized, but net turning also increased, consistent with the idea that in the absence of competition, there is greater drive biasing ipsilateral premotor circuits. Although we are not aware of evidence for a direct connection between PT clusters, we found that PT neurons project to and commissurally within the habenula, concordant with a previous report (although this earlier study found that commissural fibers originated from the migrated neurons of the PT, which is an area lateral to the cluster identified here)[51]. Thus, competition between PT clusters may take place within the habenula. In keeping with this idea, ablation of *y279* habenula neurons disrupted lateralized responses similar to PT ablation. However, ablation of the habenula commissure yielded a population of larvae with left-biased responses, similar to ablation of either right habenula or PT, possibly suggesting that the left habenula predominates in the absence of communication. Thus, although our findings do not resolve the precise mechanism by

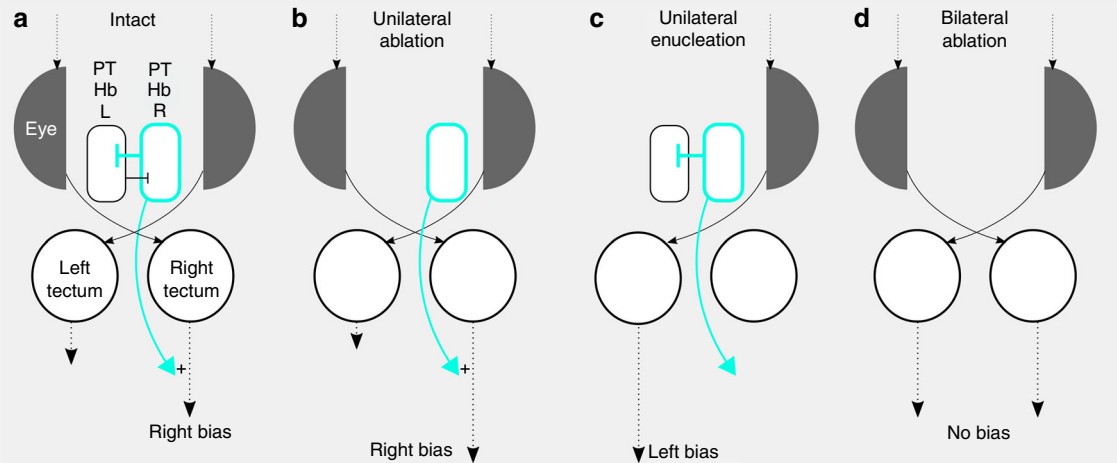

**Fig. 7 Motor asymmetry in zebrafish. a** For a right-biased larva, the dominant right PT-habenula pathway imposes right motor bias by modulating symmetric visual drive and suppressing the left PT-habenula pathway. **b** Competition between PT-habenula units is eliminated after unilateral ablations, strengthening motor bias. **c** Asymmetric visual input after unilateral enucleation overrides PT-habenula modulation of motor output. **d** Symmetric visual stimuli, after bilateral ablation of PT-habenula units, eliminate motor bias leading to randomized turn direction.

which the PT directs motor asymmetry, they demonstrate that a PT-habenula pathway is an essential substrate for lateralized behavior in zebrafish.

The last two decades have produced significant progress in understanding the morphogenetic processes that lead to consistent left–right asymmetries within the nervous system[52], and the mechanisms by which stochastic neuroanatomical asymmetry emerges when normal lateralizing cues are experimentally removed[53]. Moreover, work in *Caenorhabditis elegans* has provided important insights into the mechanisms that underlie stochastic acquisition of left–right neuronal identity[54]. In contrast, much less is known about how stochastic individual patterns of left/right behavioral identity are established. In birds, asymmetric sensory experience during a critical period in early development imprints hemispheric dominance for visual feature processing[34,55]. In contrast, we found that dark-reared larvae and *atoh7* mutants, which lack central projections of RGCs at all stages, maintained individual motor lateralization, indicating that neither spontaneous retinal activity nor asymmetric visual experience are required for individual bias to emerge. Further, as in mice and *Drosophila*, selective breeding experiments excluded the possibility that a heritable factor dictates left/right preference (Supplementary Fig. 6)[56,57]. Most likely, directional bias is stochastically determined during embryogenesis by either a specific symmetry-breaking event or by natural variability in development. While the nature of this process is not yet known, recent genome-wide association studies have provided tantalizing hints that molecular pathways involved in the development of visceral asymmetries also contribute to the establishment of human handedness[58]. Consistent with this idea, we found that left/right motor identity was disrupted by heterozygous mutations in *epb41l5*, a gene necessary for left–right patterning in mice[59]. *Epb41l5* is a membrane adapter protein[46,47] that interacts with *mind-bomb*, a ubiquitin ligase required for Notch signaling during neurogenesis[48]. Accordingly, we found that heterozygous mutations in *mind-bomb* also disrupted lateralized behavior. Surprisingly though, behavioral changes were seen in haploinsufficient mutants, suggesting that acquisition of left/right identity is highly sensitive to notch-signaling levels. Indeed, manipulations that augment or disrupt notch signaling during habenula neurogenesis can shift the production of lateral or medial neurons, and thereby isomerize the habenulae[17]. Intriguingly, *notch1a* has a similar temporal profile of expression in

the developing PT, raising the possibility that heterochronic neurogenesis of PT neuron subtypes is involved in acquisition of left/right identity[60]. Thus, although we did not observe changes in the number of *y279*-expressing neurons in the left or right PT hemisphere in *epb41l5* heterozygous mutants (Supplementary Fig. 7), future studies may reveal alterations in the differentiation of neuronal cell types associated with loss of left/right identity.

Cerebral asymmetries have been proposed to increase information-processing power through hemispheric specialization[61], or, to reflect space constraints in the brain that necessitate partitioning processing functions between hemispheres[62]. Indeed, the left and right habenula nuclei in zebrafish are activated by different sensory modalities, and have separable roles in behavior[21–24,29], and in *C. elegans* and *Drosophila*, brain asymmetries have been linked to sensory processing and memory formation, respectively[63–65]. Asymmetries in sensory-processing areas in birds are correlated with lateralized control of behavior[66]. Similarly, it seems natural to postulate that asymmetric structural or functional properties of the nervous system also underlie motor asymmetries. However, evidence for this idea is surprisingly elusive: motor lateralization has only rarely been correlated with an underlying cerebral asymmetry, and causal relationships have yet to be established[5–7,67]. The strongest relationship to date has been described in the pond snail *Lymnaea stagnalis* where the direction of coiling behavior by males during mating almost completely matches the side on which two central ganglia are fused[5].

Even less clear is the nature of the ethological advantage conferred by motor asymmetries. An enduring argument is that the hemispheres of an ideal perfectly symmetrical nervous system cannot discriminate bilaterally symmetrical sensory stimuli, and would therefore reach a stalemate in selecting a left or right motor response to a nondirectional cue[68]. Intrinsic neural asymmetries may therefore facilitate motor responses to stimuli with ambivalent directional information, avoiding simultaneous initiation of left/right responses, or reactions that are delayed due to the difficulty selecting a response direction. Indeed, during dark-induced circular swimming, larvae must initiate movement by contracting muscles on one side. In the light, even small asymmetries in the visual environment may override innate bias, explaining why we do not observe motor asymmetry under illuminated conditions. This idea is also consistent with our finding that *atoh7* mutants, which are blind, show motor bias under both

light and dark conditions; in mutants, visual stimuli do not reach the brain to override innate direction preference. We previously reported that circling behavior after loss of illumination represents a canonical local search strategy[32]. Innate motor bias may therefore expedite the initiation of light search behavior in the absence of navigational cues.

Motor asymmetries, such as human handedness, are among the most pervasive and salient forms of individual variation. Moreover, variation in motor asymmetries is linked to interindividual differences in personality, cognitive processing, and risk for neurodevelopmental disorders[14,15]. Yet, attempts to understand how such motor asymmetries are stochastically generated during development, or even their structural or functional basis in the brain, have yielded limited insight. In particular, efforts to study the development and basis for human handedness have been hampered by difficulty in identifying a neural substrate, and the inability to perform experimental manipulations. Our identification of a specific neuronal substrate for asymmetric motor behavior in zebrafish opens up a new model for understanding how functional lateralization emerges and is maintained in the nervous system. Moreover, our data also suggest that acquisition of left/right identity is sensitive to specific levels of *Notch* pathway activity. These results therefore set the stage to uncover precisely how motor bias is established during development, and is encoded by structural or functional asymmetries within the nervous system.

## Methods

**Zebrafish husbandry**. All in vivo experimental procedures were conducted according to National Institutes of Health guidelines for animal research, and were approved by the NICHD Institutional Animal Care and Use Committee. Adult zebrafish (*Danio rerio*) were maintained with a Tubingen long-fin strain background. Experiments were performed on larvae in the first 10 dpf, before sex differentiation. Larval zebrafish were raised on a 14/10-h light/dark cycle at 28 °C, at a maximum density of 20 in 10 mL of E3h medium (5 mM NaCl, 0.17 mM KCl, 0.33 mM CaCl$_2$, 0.33 mM MgSO$_4$, and 1.5 mM HEPES, pH 7.3). For dark rearing at 3 h, post-fertilization embryos were sorted into 60-mm Petri dishes at a density of 15 larvae per 10-mL medium, and placed into a dark box till 5 dpf. At 5 dpf, the media was replaced, and larvae maintained under normal light cycles till testing at 6 dpf. For controls, siblings were raised in parallel under the same conditions, except with exposure to normal 14/10 light cycles.

Transgenic lines used were enhancer traps *y279-Gal4* and *y375-Gal4*[40], *Tg(UAS: epNTR-tagRFP)y268*[69], *Tg(UAS:Kaede)s1999t*[39], *Tg(atoh7:GFP)rw021*[70], *Tg(otpb.A: Gal4-myl7:GFP)zc67*[71], and *TgBAC(vglut2a[slc17a6b]:loxP-mCherry-loxP-Gal4ff) nns21*[72]. Mutant lines were *otpa$^{m866}$*[36], *moe$^{b476}$*, and *mib$^{ta52b}$* (kind gifts of Ajay Chitnis)[45,73].

**Behavior tracking and analysis**. Behavioral tests were performed on 6–7-dpf larvae except as noted. Infrared illumination (CMVision Supplies, 850 nm) was used to monitor larvae with camera lenses fitted with IR long-pass filters to exclude visible light. Visible illumination (40 μW/cm$^2$ measured using a radiometer (International Light Technologies)), was provided by white LEDs positioned over the recording chamber (Thorlabs). Testing areas were maintained at 26–28 °C, and larvae were adapted to the recording environment for 30 min prior to starting experiments. We used DAQtimer event control software to coordinate illumination conditions and recordings[74]. When experiments required confocal imaging and behavioral analysis, larvae were raised in medium containing 200 μM PTU starting at 1 dpf. PTU was removed at least 24 h prior to behavioral recordings. Enucleations were performed at 2 dpf. Larvae were briefly anesthetized in tricaine prior to surgically removing both eyes. All steps were performed in 1× Evans physiological saline. Larvae were allowed to recover for 24 h in Evans before returning to E3h media. Experiments were performed at 6–7 dpf. Controls were identically anesthetized and temporarily reared in 1× Evans.

*Trajectory analysis:* For path trajectory analysis, images were captured with a uEye IDS1545LE-M CMOS camera (first Vision), and larvae tracked in real time using DAQtimer[32]. Individual larvae were placed into a 120 × 120-mm arena. Each individual was tracked for 30 s after loss of illumination over 4 successive trials separated by 3 min of baseline illumination (with the same protocol used for constant illumination trials, except the light that remained on throughout). Where behavioral experiments were coupled with manipulations (chemogenetic or laser ablation, genetic mutations), baseline controls were obtained by recording the 30-s illuminated interval prior to the light-off interval. We used three measures to characterize directional bias: NTA, Absolute NTA, and Match Index. *NTA:* the sum total of all leftward (− degrees) and rightward (+ degrees) path direction changes

over a 30-s recording interval. Thus, individuals that changed direction equally to the left and right would have an NTA of 0, indicating no net directional preference. *Absolute NTA:* as for the NTA but taking the absolute value of each path direction change before summation. The absolute NTA measures the total amount of turning behavior during a 30-s interval. *Match Index:* Used for experiments where larvae were subjected to four trials; the Match Index is the fraction of trials 2–4 where the NTA had the same sign as on the first trial. We excluded trials where we obtained less than 10 s of data for an individual. A MI of 1.0 indicates that a larvae turned in the same direction on trials 2–4, as on trial 1, whereas a MI of 0.33 would occur if only one of trials 2–4 was performed in the same direction as on trial 1.

*Kinematic analysis:* To measure routine-turn initiation, we used a high-speed camera (DRS Lightning RDT/1, DEL Imaging) at 1000 Hz with off-line analysis of video images using Flote[75]. Larvae were tested in a 58 × 58-mm arena with IR illumination, and recordings were triggered when larvae entered a centrally placed 15 × 15-mm region of interest (ROI) using DAQtimer. To minimize environmental visual cues that might influence trajectories, the arena was completely enclosed with a diffuser underneath the arena and a fitted lid above the arena holding a IR long-pass filter. We performed 4 recordings (10-s duration) after loss of illumination separated by 3 min of baseline illumination. Control experiments were the same, but without the dark periods. We analyzed only larvae that executed at least three routine turns on each of the four trials.

*Startle direction assay:* Acoustic startle tests were performed using DAQtimer to send 3-ms duration, 1000-Hz sinusoidal pulses to an electrodynamic exciter (Type 4810 Mini-shaker; Brüel & Kjær) via a digital-analog data acquisition board (PCI-6221, National Instruments). Responses were captured using the high-speed camera and analyzed using Flote[76]. To determine if startle direction bias correlated with dark-circling direction bias, larvae were classified as left/right at 6 dpf using trajectory analysis over 4 trials. At 7 dpf, larvae were placed into a 3 × 3 grid of 1 × 1-cm wells. For trials in the dark, larvae were tested with 16 repeats of an auditory/vibrational stimulus (~18 dB relative to 1 m/s$^2$), occurring 10 s after loss of illumination. Each repeat was separated by 2 min of constant illumination. Trials in the light were the same, without the 10-s dark period. Only larvae performing at least four long- or short-latency startle responses were analyzed. For analysis, we compared the percentage of short- and long-latency C starts performed in a rightward direction for each preclassified group of larvae. Startle in unilaterally ablated PT larvae was performed as described above, with the exception that startle was performed at 7 dpf, allowing sufficient recovery from PTU for accurate tracking and kinematic analysis.

*Two-target phototaxis assay:* For the two-target phototaxis assay, we first identified the left/right identity of larvae at 6 dpf using trajectory analysis over four dark trials. Larvae with consistent directional responses to all four trials were then retained for testing the next day. At 7 dpf, these larvae were individually placed into a 58 × 58-mm arena and given 5 min to adapt to baseline illumination (20 μW/cm$^2$). After adaption, a 15 ×15-mm ROI at the center of the arena was monitored by DAQtimer. Once the larva entered the ROI, full-field illumination was extinguished, and real-time tracking of the position and orientation of the larva was started. After 3 s, two equal-intensity light spots (20 μW/cm$^2$, 6-mm radius) were positioned 10 mm from the head of the larva at 55° from its current orientation. Light spots were projected onto the base of the arena (AAXA P2 Pico Projector). At the same time, a high-speed camera was triggered to capture the response of the larva to the phototaxis stimuli. Each larva was tested four times with 5 min of illumination between trials. Trials were excluded from analysis if a light spot was obscured by the perimeter, or mispositioned due to sudden movement of the larva. Only larvae with at least two trials were analyzed.

*Multiday bias persistence tests:* (1) For testing individual motor asymmetry over 24 h, we first classified individuals as left- or right-biased at 6 dpf using kinematic routine-turn analysis as described above. Classification was determined using the directional preference on the first trial only. Larvae were individually housed and returned to the incubator overnight. We then retested larvae at 7 dpf, using the same assay. (2) To determine if motor asymmetry persisted over longer timelines, larvae were tested at 6 dpf using path trajectory analysis over four dark trials. Individuals performing same-direction circling for all four trials were were then individually housed, and starting at 7 dpf fed daily (AP100 larvae dry food, Zeigler) with the media partially replaced 3 h after each feeding. No food was given on the day of testing. At 10 dpf larvae were retested using the same assay.

*Heritability analysis:* We selectively raised larvae classified as left- or right-biased over two generations. In each generation, parents were incrossed, and larvae tested using kinematic analysis over four trials. To enrich for individuals with consistent motor asymmetry, we raised only larvae that (1) had a percentage rightward-turn use in the top or bottom quartile of responses, and (2) performed all four trials in the same direction. For each generation, left-/right-classified adults were incrossed as groups, and a minimum of three independent clutches combined for analysis.

**Imaging**. *Calcium imaging:* We co-injected 50 ng of *UAS:nls-GCaMP6s* plasmid (derived from *UAS:nls-GCaMP6s-2a-nls-dsRed*[77]) with 80 ng of tol1 RNA into one-cell-stage *y279:Gal4* embryos, and raised in medium containing 200 μM PTU. At 6 dpf, larvae were mounted in 2% low-melting-temperature agarose, and imaged using a 20× immersion objective on an upright Leica TCS-SP5II microscope. To avoid visual stimulation during GCaMP imaging, we used a 2-photon Spectra-

Physics MaiTai DeepSee laser tuned to 950 nm, and installed a 620-nm LED (All Electronics) on the stage to provide visual stimulation at 80 μW/cm². We performed three trials per larva separated by 3 min of constant red light illumination, with each trial consisting of 60 s of light, and 60 s of dark. We captured a single plane through the PT using 16× line averaging at ~0.96 Hz. After imaging, larvae were placed in fresh E3h media overnight. At 7 dpf, we used trajectory analysis of dark-induced circling to determine left/right identity. We analyzed fluorescence intensity changes by manually outlining an ROI around each neuron in a maximum projection of the time-series data, and calculated $(F_t − F_0)/F_0$ ($\Delta F/F$) where $F_0$ was the mean fluorescence intensity during the first baseline period. Neurons with a $\Delta F/F$ greater than $3S$ over three successive time points were classified as responders, where S was the standard deviation of $\Delta F/F$ during the first baseline period.

*Photoconversion:* We characterized projections from *y279-Gal4, UAS:Kaede*-expressing rostral PT neurons by focal photoconversion of Kaede from green to red fluorescence in one hemisphere. At 2 dpf, PT neurons are a salient bilateral cluster in the ventral diencephalon. Larvae were mounted in 2% low-melting-temperature agarose on an upright Leica TCS-SP5II confocal microscope. We used a 25× immersion objective, and set the field of view to comprise 1–2 Kaede-expressing PT neurons, imaged with a 488-nm laser. For photoconversion, we performed a single scan with 8-frame averaging using a 405-nm laser at 5% power. Conversion of Kaede to the red fluorescent state was confirmed by imaging with a 568-nm laser. After photoconversion, embryos were removed from agarose, and maintained in E3h until 5–6 dpf for imaging. To assess the spatial relationship between RGC projections and rostral PT neurons, we globally photoconverted Kaede in *y279-Gal4, UAS:Kaede,* and *atoh7:GFP* larvae using a 9-W 405-nm LED for 20 min (Formlabs, Form Cure). To quantify habenular neuropil derived from *y279* PT projections, confocal z stacks in photoconverted larvae were captured using matched imaging conditions spanning the entire habenula volume. Photoconverted habenula neuropil area was traced in maximum projections and quantified in ImageJ.

*Neuron counts:* We used Imaris to count *y279-Gal4, UAS:Kaede*-expressing neurons in the rostral PT in larvae imaged at 6 dpf. After imaging, larvae were extracted from the agar, individually maintained in 6-well plates for 24 h, and then tested at 7 dpf to determine left/right identity using path trajectory analysis of dark-induced circling behavior.

**Ablations.** *Genetic ablations:* For nitroreductase ablations, Gal4 lines were crossed to *UAS:epNTR-tagRFP,* and RFP⁺ embryos were raised to maturity. Carriers were incrossed, and embryos were sorted using epifluorescence at 2–3 dpf into NTR⁺ red fluorescence or NTR⁻ groups. Each group was treated with 7.5 mM metronidazole in E3h media from 3 to 5 dpf. Metronidazole media was refreshed after 24 h. At 5 dpf, larvae were returned to fresh E3h media. At 6 dpf, genetically ablated larvae were tested, and a subset imaged on a confocal microscope to validate ablation efficacy.

*Laser ablations:* We immobilized *y279-Gal4;UAS:Kaede* larvae with tricaine, and performed laser ablation using a Spectra-Physics MaiTai DeepSee laser (tuned to 800 nm, beam power 2 W) on an upright Leica TCS-SP5II microscope with a 20× immersion objective. For ablation, we set the ROI to 1–2 neurons per field of view. Efficient ablation was accompanied by the acute and transient development of a bubble. Controls were anesthetized and mounted for the same duration as ablated larvae. We performed path trajectory analysis during four dark-induced circling trials per larva. For bilaterally ablated larvae, we classified left/right identity using the first trial, then calculated the mean NTA for trials 2–4. For unilaterally ablated larvae, where we aimed to see if ablation imposed a pattern of lateralized behavior, we took the mean NTA for all four trials. After behavioral analysis, we imaged the ablated region to assess ablation efficiency, and analyzed only larvae with near-complete absence of targeted neurons (~50% of ablated individuals).

**Genetic mapping.** To map *moe^y606*, we formed mutant and sibling pools, each consisting of 75 embryos that were derived from a single clutch. We Trizol-extracted total mRNA, purified on a Qiagen RNA mini-cleanup column, and sequenced samples using single-end 100-bp reads using an Illumina HiSeq (100 M reads per sample). We then used RNAmapper in Galaxy (galaxyproject.org) to perform bulk segregant analysis, identifying a critical region on chromosome 9 and a 7.7-fold reduction in expression of *epb41l5* within that interval. We noted that no reads were present in exon 1 or 2 of *epb41l5*; these exons amplified correctly from sibling genomic DNA but not from mutant DNA (exon 1 primers: 5′-TCCAC TTTTGGGGATTTACG, 5′-ATTCAATGGCGGAGCAATAC; exon 2 primers: 5′-GGCCATTGACAGTAGTGTGG, 5′-TGACAAGACGCTGAACAAGC), suggesting the presence of a deletion. We then used PCR to assess the presence of genomic DNA in mutants in intervals across a 20-kbp region between exon 3 of *epb41l5* to the 3′UTR of the neighboring gene *ptpn4a*. This narrowed the candidate interval to 7 kb, allowing us to design primers to amplify and sequence across the deletion, revealing loss of genomic DNA from chr9:213682 to 218117 (GRCz11, chr9_KZ114909v1_alt). In complementation testing with *moe^b476*, we recovered the curly-up phenotype in 50/192 (26%) embryos, confirming that this was the causative mutation. Sequencing of the PCR product that spanned the deletion enabled us to design a genotyping protocol to distinguish wild-type and heterozygous larvae using 3 primers (5′-CTACCTGAACAAACTCAATCCAGTC, 5′-AACCATAATAAAATGAGCGTCTCT, and 5′-TCATTTTGAAATGCCTGCAA).

These primers amplify a single 296-bp band from the wild-type locus, and a 333-bp band from embryos with the deletion.

Previous work established the presence of a large deletion in *moe^b476*, but did not define its precise boundaries, precluding genotyping of heterozygotes[45]. To map the deletion, we used whole-genome shotgun sequencing with 100-bp paired-end reads. After mapping reads to GRCz11 with bowtie2, we focused on a large read-depleted region on chr9 covering *epb41l5*. IGVviewer revealed five reads where the paired-end reads spanned the read-depleted region. PCR across the putative deletion using primers 5-AAACTGCATAAGTGCCTCACC and 5-GAGACATCGATTCCGCTTTG amplified an ~600-bp band, which sequencing confirmed had the expected genomic sequence at each end, and demonstrated that the deletion in *moe^b476* spans chr9:28835868 to 29370835, completely deleting *epb41l5, ptpn4a, tmem177,* and *pth2r,* and exons from *zgc:91818* and *hs6st3b*. We also used these primers to genotype *moe^b476* outcrosses because only heterozygous larvae yielded the 600-bp band.

*Genotyping:* After behavioral analysis, we genotyped *moe^y606* and *moe^b476* larvae as described above. *mib^ta52b* larvae were genotyped by PCR amplification across the mutation (primers: 5′ GGTGTGTCTGGATCGTCTGAAGAAC; 5′ GATGGATGT GGTAACACTGATGACTC, product size 194 bp). To discriminate wild-type from heterozygous larvae, PCR products were digested with the restriction enzyme NlaIII (N.E.B.) that digests wild-type but not mutant product (WT 155 bp; Het 155, 194 bp; Mutant 194 bp). For experiments with otpa mutants, we compared the "cousin" offspring of wild-type (+/+) and mutant (−/−) sibling parents.

**Statistical analysis.** Analysis was performed in IDL (Harris), RStudio (Mathworks), Gnumeric (http://projects.gnome.org/gnumeric/), and JASP (https://jasp-stats.org/). Data in the figures and text are means ± SEM except as noted. All *t* tests were two-sided. Box plots show median and quartiles with whiskers indicating 10–90%, and red diamonds indicating means. Normality was determined by the Shapiro–Francia test. Analysis of non-normal data sets was performed using the Mann–Whitney *U* test, or for one-sample comparisons to a given number, through Monte Carlo testing (see below). Effect sizes reported in this are Cohen's *d* for *t* tests, the partial ETA-squared ($\eta^2_p$) for ANOVAs, and the rank-biserial correlation *r* for Mann–Whitney *U* tests.

To analyze whether distributions of turn direction deviated from that which would be expected if larvae did not show direction bias, we used Monte Carlo simulations rather than comparison with a binomial distribution because the direction of sequential turns is not independent[32,78]. In each simulation, we used the same number of larvae as in the experimental data (i.e., 89 and 39, respectively for light-off and baseline conditions), and the same number of routine turns produced on each of the four trials for each larva. As sequential routine turns show a statistically significant likelihood to be executed in the same direction, we simulated routine-turn direction across events on a given trial by randomly selecting a turn direction for the first event, then using the previously measured "lock index" to weight a random decision on whether each subsequent turn would be in the same or reverse direction. We used lock indices of 62.8 and 13.2 for light-off and baseline simulations, respectively (corresponding to 81.4 and 56.6% chance of executing sequential same-direction routine turns)[32]. The expected histograms of % rightward turns in Fig. 1 are derived from 10,000 simulations. In the simulation data, 10% of larvae showed less than 23% rightward turns or greater than 76% rightward turns during light-off conditions (i.e., a modal value of 9 of the 89 larvae), whereas in our data, 37 of the 89 larvae exceeded these thresholds. This proportion was greater than all 10,000 simulations; hence, we assign a *p* value of <0.0001. In comparison, under baseline condition simulations, 10% of larvae showed less than 31% rightward turns or greater than 68% rightward turns during light-off conditions, and in our data 7 of the 39 larvae exceeded these thresholds. However, this proportion was exceeded by 1195 simulations; hence, we assign a *p* value 0.119. We used a similar procedure (one-sample permutation test) to compare the means of larval Match Indices to 0.5, by ranking the difference between the mean of actual match indices in experimental data, with means derived from 100,000 simulations, using the same number of larvae and trials per larva under the null hypothesis assumption that larvae had a 50% (i.e., random) probability of matching the trial 1 direction on each of trials 2–4.

**Reporting summary.** Further information on research design is available in the Nature Research Reporting Summary linked to this article.

## Data availability
Gal4 lines are available from the Zebrafish International Resource Center (http://zebrafish.org). Further information and requests for image datasets and analysis software should be directed to Harold Burgess (burgessha@mail.nih.gov) or Eric Horstick (eric.horstick@mail.wvu.edu). Numerical data underlying Fig. 1b–e, g–j, 2c–f, 3b, g–i, 4c–e, 5e–g, 6c–h, and Supplementary Figs. 1b, i–j, 2a, f–h, 3e–f, g, i–j, 6 and 7 are provided in a Source Data file.

## Code availability
The analysis codes that were used in this study are available from the corresponding author upon request.

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

## Acknowledgements
We thank Jennifer Sinclair for expert technical support, Dr. Katie Drerup for assistance with RNAmapper, John Hageter for assistance with calcium imaging analysis, Greg Palardy for the mib genotyping protocol, and the NICHD Molecular Genomics Core for library preparation and sequencing. This work was supported by the Intramural Research Program of the Eunice Kennedy Shriver National Institute for Child Health and Human Development (NICHD) and utilized the high-performance computational capabilities of the Biowulf Linux cluster at the National Institutes of Health, Bethesda, MD.

## Author contributions
E.J.H. and H.A.B. conceived the experiments and wrote the paper. E.J.H., Y.B., and H.A.B. performed the experiments. All authors approved the final paper.

## Competing interests
The authors declare no competing interests.
