## [Peer Review File · Nature Communications]

Reviewers' comments:

Reviewer #1 (Remarks to the Author):

This is a very interesting manuscript exploring the mechanisms driving motor asymmetry in larval zebrafish. The authors find that upon loss of illumination, zebrafish larvae display preferential turning to one direction, persistently across trials and over days. This preference among individual larvae for directional bias is also present in a visual spot choice assay and in response to acoustic startles in the dark. They then carried out a circuit-breaking screen to determine that a group of neurons in the Posterior Tuberculum (PT) are involved in imparting motor asymmetry. Elegant laser-guided ablation experiments confirmed these findings. Using functional imaging, the authors show that some neurons in PT are responsive to visual stimuli but there is not an obvious bias between hemispheres. Photoconversion of neurons in the PT labelled neurites that cross the midline and also arborize in the habenular nuclei. This suggests a role for a PT-Habenula (Hb) pathway in motor asymmetry. Finally, genetic disruption of the Notch pathway replicated motor asymmetry phenotypes as those observed by ablation of the PT.

Overall, the authors present some very interesting results related to bias in motor output that varies among individual larvae. The behavioral assays and ablations are very carefully thought through and are convincing. The study provides convincing evidence of motor asymmetry and an involvement of the PT in generating this motor asymmetry. However, we believe that additional experiments could potentially clarify the role for the PT-Hb pathway in generating directional behavioural bias. We also consider that the data on the Notch pathway mutant adds little to the study.

(1) The authors claim that motor asymmetry is only evident in the absence of visual information. However, loss of illumination can also be considered a stimulus for visual circuitry and the two-target choice assay is also dependent on visual input. Consequently the role of the PT-Hb in eliciting asymmetric motor behaviours could be dependent on visual circuit-triggered events or could be independent of the sensory modality triggering the behaviour. This could presumably be assessed if the role of the PT-Hb pathway is assessed in all the behaviours that elicit motor asymmetries. We suggest bilateral and unilateral ablations of PT (for pre-sorted larvae with left vs right bias in the circling assay), followed by testing in the two-spot choice and acoustic startle assays. We believe these experiments would help the authors disentangle whether the PT neurons contribute to motor asymmetry with respect to a specific sensory modality or more generally.

(2) It is unclear from the photo-activated labelling of PT projections to the Hb whether these are symmetric to both the left and right hemisphere. Published data of visually sensitive and/or forebrain nuclei projecting to the Hb are confusing but some suggest asymmetric projections to the left Hb (see, for example, Turner KJ et al., *Frontiers in Neural Circuits*, 2016; Cheng RK et al, *BMC Biology*, 2017 and Zhang B et al, *Neuron*, 2017)). Figure 5, panels C and D suggest more labelling in the left Hb neuropil than right Hb – is this correct and can the authors quantify the innervation? Do PT neurons form a left Hb asymmetric visual pathway or a symmetric one? The Ca²⁺ imaging data in Figure 4 suggests that PT-Hb might be a visually driven pathway. The authors should discuss whether the pathway they describe is likely the same or different from diencephalon to habenula projections described in previous reports.

(3) The authors present some Ca²⁺ imaging data which suggests that PT neurons are ON and OFF responsive to visual stimuli. However, they find no bias between left and right PT nuclei. We believe that further analyses could clarify how PT activity relates to behaviour (even though the circling

behaviour cannot be elicited in tethered larvae upon loss of illumination). Ca²⁺ imaging in PT and Hb after pre-sorting the larvae for motor lateralisation could reveal insights into how left and right neurons respond differently (or not) to visual stimuli in left or right lateralised fish. Additionally, the authors should extend the duration of the OFF pulse beyond the 57 second window to determine whether PT neurons are simply sustained OFF and ON responsive or does their activity taper off at time-scales corresponding to the circling behaviour?

(4) It is understandable that the authors have included analysis of the *moe* (and *mib*) mutant(s) given that the behaviour is disrupted in the mutant(s). However the analyses of the mutant don't reveal any significant insights into the development or function of the circuitry underlying motor asymmetry. The Notch pathway is involved in generation of all neurons and as yet there is no phenotypic analysis to link the mutant phenotype directly to the circuit described. Consequently several statements like “developmental pathways that establish visceral asymmetries also govern acquisition of left/right identity” have little value. The advantage of including the mutants is that it alerts the community to useful behavioural mutants but it adds little to the narrative of this study. If the authors decide to keep the mutant description in the paper, I would minimise it and remove the more speculative text.

Minor issues:

(1) It would be of interest to some in the community if the authors were to show distributions of orientation angles during circling behaviour in left vs right biased larvae.

(2) The closed-loop two-target assay is a very nice experiment; but the loss of illumination before the test does not come across clearly to the reader in the results.

(3) Figure 3(b), middle panel might have an error. The control and ablated group Match Index during darkness is shown to be non-significant but the results indicate this is a significant difference.

(4) Why use the terminology “epithalamic commissure” – is this different from habenular commissure?

(5) Do the habenular ablations leave the PT commissural projections intact?

(6) Line 301: “In contrast, much less is known about how stochastic individual patterns of left/right identity are established”. True, but there are some good examples of this in worm and indeed in the fish habenulae, if the overriding influence of left-sided Nodal signalling is removed, it is proposed that stochastic differences in Fgf signalling can determine handedness.

Reviewer #2 (Remarks to the Author):

In this paper, Horstick and colleagues report the interesting observation that larval zebrafish display lateralized motor behavior upon sudden loss of illumination. They use this observation to identify neurons that influence lateralized motor behavior. Specifically, by lesioning random subsets of neurons using GAL4 lines, they suggest that neurons in the posterior tuberculum and habenula are part of a circuit influencing lateralized behavior. This is complemented by calcium imaging and laser ablation experiments. The authors additionally use genetics to identify a signaling pathway that influences motor asymmetry, although it is unclear how this is linked to the structures defined earlier.

There are a number of points that should be addressed to increase the impact of the paper.

1. The cells in the posterior tuberculum respond to light. Is there a direct retinal innervation to the posterior tuberculum? It would be interesting to inject a lipophilic tracer into the retina to see if there is direct connectivity with the cells in $\gamma 279$ or $\gamma 375$. How does the projection fit with previous reports e.g. Robles et al¹ and Easter and Burrill². In other words, what AF would this be?
2. It should be noted somewhere in the discussion whether there are known homologies between the posterior tuberculum and brain regions in other vertebrates such as mammals. This would help place this work in a broader context. Information on connectivity (e.g. input from the retina) would help in this respect.
3. How does this work fit in with the response to loom described by Heap et al³? This should be considered in the discussion, especially with regards to lines 272-274. Do the PT neurons here project to the tectum?
4. Where possible, effect sizes should be indicated. In comparing behavior of groups, plotting differences would be an effective way of presenting the data. Individual data points should be shown, e.g. in a jitter plot, where possible. (e.g. Fig. 1e)
5. The commissure running between left and right habenula is the habenular commissure, not the epithalamic commissure.
6. What is the significance of the vGluT label in Fig. 3f? Are $\gamma 279$ cells neither glutamatergic nor GABAergic (Fig. S2)? Before excluding the possibility that the cells are GABAergic, direct imaging of a double transgenic (eg $\gamma 279$ driving GFP and *gad1b:DsRed*) should be carried out.

The following minor errors should be corrected:

1. Line 317: the phrase that mindbomb is “a ubiquitin ligase required for notch signaling” is already stated (line 315-316).
2. Line 481: the word “line” should be “objective” or “lens”
3. Line 484: “Fb” is undefined.
4. Line 771: In figure 6c, homozygous is on the left and heterozygous is on the right.

References

1. Robles, E., Laurell, E. & Baier, H. The retinal projectome reveals brain-area-specific visual representations generated by ganglion cell diversity. *Curr Biol* 24, 2085–2096 (2014).
2. Burrill, J. D. & Easter, S. S. Development of the retinofugal projections in the embryonic and larval zebrafish (*Brachydanio rerio*). *J Comp Neurol* 346, 583–600 (1994).
3. Heap, L. A. L., Vanwalleghem, G., Thompson, A. W., Favre-Bulle, I. A. & Scott, E. K. Luminance Changes Drive Directional Startle through a Thalamic Pathway. *Neuron* 99, 293–301.e4 (2018).

Reviewer #3 (Remarks to the Author):

In the manuscript, the authors described three types of motor asymmetries of larval zebrafish, including dark-induced circular swimming, two-alternative phototaxis, and acoustic startle behavior. They found that, for each larva, the left/right direction preference was consistent across the three types of behavior, sustained across contiguous trials and across several days, but was overridden by visual inputs. They also addressed the ontogeny of the asymmetry, which was neither dependent on

the earlier visual experience nor inherited. Taken together, the authors characterized a robust motor asymmetry. Furthermore, the authors tried to examine the circuitry and molecular relevance of such asymmetry, including the PT-habenula pathway and mutations like *moe* and *mib*, but the results were not clear. Taken together, this is an interesting and somehow preliminary work on motor asymmetry and can be improved by well-designed mechanism studies.

Major concerns:

1. As the authors have shown in Fig. 1g, the left-preferring and right-preferring larvae were found with similar proportions. The authors may want to examine the individual difference between the left-preferring and the right-preferring larvae, to address the underlying mechanisms either on circuit or on molecular signaling, instead of comparing the preference distribution between the normal and the genetically modified population. It is important to classify the larvae according to their performance in the circular swimming and then examine their tubercle and habenula dark responses and their PT-habenula connection.
2. It had better provide information on how many Gal4 lines were screened and what brain areas were covered in these lines. Are there any other lines labeling anterior PT or affecting the motor asymmetry? Especially, as ablation of bilateral habenulae abolished the asymmetry, why such defect was not found on some Gal4 lines labeling the habenulae?
3. In Fig. 3b, the authors showed that the difference in Match Index between $\gamma 375$ and control was not significant. Thus they should provide rationale on describing 'a decrease in motor bias after ablation'?
4. The authors found that motor bias was decreased in two Gal4 lines, $\gamma 279$ and $\gamma 375$. The authors then postulated that neurons affected in both lines should be responsible for the motor bias. This may have missed other candidates, as the non-overlapping neurons in the two lines may reside in the same pathway innervating the motor bias effector, or they may work synergetically in other manner.
5. Although $\gamma 279$ ablation reduced the motor asymmetry, it is possible that anterior PT neurons may just respond to the dark signal unilaterally and initiate the turning motor behavior. This can explain both that these neurons area necessary and that the no left/right difference was found in their activities. Besides, it would be better if the authors can show acoustic startle assay results or two-target phototaxis assay results of these larvae.
6. Alternatively, the PT neurons may compete or their downstream neurons, like in the habenulae, may compete in driving asymmetric behavior. It may be quite informative to examine such competition and compare between the left- and the right- preferring larvae, like figuring out the PT-habenula connection. Besides, it would be better if the authors can free the tail or record electromyography (EMG) or do calcium imaging on motor neuronal activities on the restrained larvae, to examine the neurons contributing to the generation of motor asymmetry.
7. The authors found weak motor bias on the $\gamma 606$ mutant. While this line may be quite informative in examining the asymmetry mechanism beyond the morphological abnormality, the authors went on to examine its relationship to the Nodal pathway. As the Nodal pathway affects left-right asymmetry development, which was not found in previous left- or right- preferring larvae, such experiments lack sufficient rationale.

Minor concerns:

1. What's the ecological significance of the motor asymmetry found in dark but not in light?

2. In determining the visual contribution, the authors employed *atoh7* mutants. While RGB inputs were abolished in this line, there is also possible that the deficiency in other brain areas expressing *atoh7*, like hindbrain nuclei, played their roles.
3. It deserves examination how the visual information predominates over intrinsic motor asymmetry.
4. For larvae tested at 7 or 10 dpf, there were some larvae which changed their direction preference (Fig. 1j). How such did changes happen?
5. In the reference session, the journal names were in abbreviation, like 67, or in full , like 68. The authors may use a consistent style.

This is a very interesting manuscript exploring the mechanisms driving motor asymmetry in larval zebrafish. The authors find that upon loss of illumination, zebrafish larvae display preferential turning to one direction, persistently across trials and over days. This preference among individual larvae for directional bias is also present in a visual spot choice assay and in response to acoustic startles in the dark. They then carried out a circuit-breaking screen to determine that a group of neurons in the Posterior Tuberculum (PT) are involved in imparting motor asymmetry. Elegant laser-guided ablation experiments confirmed these findings. Using functional imaging, the authors show that some neurons in PT are responsive to visual stimuli but there is not an obvious bias between hemispheres. Photoconversion of neurons in the PT labelled neurites that cross the midline and also arborize in the habenular nuclei. This suggests a role for a PT-Habenula (Hb) pathway in motor asymmetry. Finally, genetic disruption of the Notch pathway replicated motor asymmetry phenotypes as those observed by ablation of the PT.

Overall, the authors present some very interesting results related to bias in motor output that varies among individual larvae. The behavioral assays and ablations are very carefully thought through and are convincing. The study provides convincing evidence of motor asymmetry and an involvement of the PT in generating this motor asymmetry. However, we believe that additional experiments could potentially clarify the role for the PT-Hb pathway in generating directional behavioural bias. We also consider that the data on the Notch pathway mutant adds little to the study.

We appreciate the reviewer's enthusiasm, and careful reading of our manuscript, and the excellent ideas for additional experiments, which we have tried to respond to in full.

(1) The authors claim that motor asymmetry is only evident in the absence of visual information. However, loss of illumination can also be considered a stimulus for visual circuitry and the two-target choice assay is also dependent on visual input. Consequently the role of the PT-Hb in eliciting asymmetric motor behaviours could be dependent on visual circuit-triggered events or could be independent of the sensory modality triggering the behaviour. This could presumably be assessed if the role of the PT-Hb pathway is assessed in all the behaviours that elicit motor asymmetries. We suggest bilateral and unilateral ablations of PT (for pre-sorted larvae with left vs right bias in the circling assay), followed by testing in the two-spot choice and acoustic startle assays. We believe these experiments would help the authors disentangle whether the PT neurons contribute to motor asymmetry with respect to a specific sensory modality or more generally.

We agree: loss of illumination is clearly a trigger for the movement occurring during circling behavior, and have removed the Line in the discussion that read, "Lateralized motor behavior occurs both during spontaneous movement, and in response to visual and auditory cues."

Also based on the larval strategy for phototaxis (i.e. turn away from the eye that has a greater reduction in illumination - Burgess, *Curr Biol* (2010) the two-target choice assay is quite possibly just a special case of turning after loss of illumination and accordingly we have removed the expression 'three assays' in the results section.

As requested, we performed a new experiment where we ablated PT neurons prior to measuring dark induced startle direction (however, we did not pre-determine bias - we need to allow sufficient time for recovery and ablations are done prior to bias onset). We found that as for the circling behavior, unilateral PT ablation dictates bias in a very similar way to the effect of ablation on circling behavior (Fig. 3h). We have added this data to Figure 3i and added the text to the results.

(2) It is unclear from the photo-activated labelling of PT projections to the Hb whether these are symmetric to both the left and right hemisphere. Published data of visually sensitive and/or forebrain nuclei projecting to the

Hb are confusing but some suggest asymmetric projections to the left Hb (see, for example, Turner KJ et al., *Frontiers in Neural Circuits*, 2016; Cheng RK et al, *BMC Biology*, 2017 and Zhang B et al, *Neuron*, 2017)). Figure 5, panels C and D suggest more labelling in the left Hb neuropil than right Hb – is this correct and can the authors quantify the innervation? Do PT neurons form a left Hb asymmetric visual pathway or a symmetric one? The Ca²⁺ imaging data in Figure 4 suggests that PT-Hb might be a visually driven pathway. The authors should discuss whether the pathway they describe is likely the same or different from diencephalon to habenula projections described in previous reports.

To resolve this question, we have now performed unilateral PT photoconversions in identified left/right biased larvae, then measured the area of habenular neuropil arising from PT projections in both hemispheres. This data is included as Figure S3e-f and description to the results "PT projections terminated in the neuropil region of both habenula hemispheres with a significant majority terminating at the left habenula regardless of PT converted (Figure S3e) or motor bias (Figure S3f)".

Of previously identified habenula projections, the neurons we identify are caudal to the ventral entopeduncular nucleus and the thalamic eminence, the two most likely candidates among areas described in papers mentioned by the Reviewer. We cited Hendricks and Jesuthasan *J Comp Neurol* (2007), because that work identified a bilateral PT projection to the Habenula. However there is a discrepancy with our study, as Hendricks *et al* suggested the origin is in migrated neurons of the PT: these are lateral to the neurons we identified here. We have added this detail to the Discussion.

(3) The authors present some Ca²⁺ imaging data which suggests that PT neurons are ON and OFF responsive to visual stimuli. However, they find no bias between left and right PT nuclei. We believe that further analyses could clarify how PT activity relates to behaviour (even though the circling behaviour cannot be elicited in tethered larvae upon loss of illumination). Ca²⁺ imaging in PT and Hb after pre-sorting the larvae for motor lateralisation could reveal insights into how left and right neurons respond differently (or not) to visual stimuli in left or right lateralised fish. Additionally, the authors should extend the duration of the OFF pulse beyond the 57 second window to determine whether PT neurons are simply sustained OFF and ON responsive or does their activity taper off at time-scales corresponding to the circling behaviour?

The calcium imaging in Fig. 4 was already in fish with identified motor bias, and we did not observe a difference. To increase statistical power we performed additional PT calcium imaging experiments – adding 71 neurons that we incorporated into Figure 4e and still do not see a difference in activity between left/right PT in left/right biased larvae. A difficulty that we can not easily overcome, is that embedded larvae do not produce biased tail movements. The lack of asymmetric activity may reflect the recording conditions, so we did not make any claims in the text about the meaning of this result, beyond that PT neurons respond to loss of illumination.

As requested, we also recorded PT neuron activity over a longer period (3 minutes of dark) and added this as a new panel Fig 4d. We added text to the Results describing the match between activity and behavior: "Consistent with the time-frame of circling behavior (Figure 1b) more than half (27/46) of the OFF responsive neurons returned to half-maximal activity within 30 seconds of the light OFF stimulus, with a mean time from the OFF stimulus to half max activity of $22.4 \text{ s} \pm 14 \text{ s}$ (mean/std. dev.). Interestingly, activity remained elevated above baseline thereafter, even during prolonged dark periods, possibly accounting for the lower level of motor bias seen for several minutes after loss of illumination (Figure 4d)."

We also imaged responses in habenula neurons, and added the text in the results, "Consistent with reports that the habenula is activated by both light ON and OFF cues, calcium imaging revealed a

subset (16/101) of *y279-Gal4* neurons that responded during light ON/OFF transitions (Figure S3j).". In the figure legend we note that only 1 right habenula neuron had an OFF response excluding additional analysis between hemispheres.

(4) It is understandable that the authors have included analysis of the *moe* (and *mib*) mutant(s) given that the behaviour is disrupted in the mutant(s). However the analyses of the mutant don't reveal any significant insights into the development or function of the circuitry underlying motor asymmetry. The Notch pathway is involved in generation of all neurons and as yet there is no phenotypic analysis to link the mutant phenotype directly to the circuit described. Consequently several statements like "developmental pathways that establish visceral asymmetries also govern acquisition of left/right identity" have little value. The advantage of including the mutants is that it alerts the community to useful behavioural mutants but it adds little to the narrative of this study. If the authors decide to keep the mutant description in the paper, I would minimize it and remove the more speculative text.

We removed the statement about the relationship to visceral asymmetries from the abstract and introduction. We have retained the mutants and mention of this idea in the discussion because we think that it is very cool that overtly normal heterozygous mutants lack motor-bias. We suspect that similar gene dosage effects will be critical for motor bias in other species and would like to introduce this idea.

Minor issues:

(1) It would be of interest to some in the community if the authors were to show distributions of orientation angles during circling behaviour in left vs right biased larvae.

We added new data, Figure S1d, showing the change in orientation over time for left and right identified larvae during both baseline and dark responses.

(2) The closed-loop two-target assay is a very nice experiment; but the loss of illumination before the test does not come across clearly to the reader in the results.

We added the text "after the loss of full field illumination" in the Results section where we first describe this experiment.

3) Figure 3(b), middle panel might have an error. The control and ablated group Match Index during darkness is shown to be non-significant but the results indicate this is a significant difference.

Thank you, the figure was wrong, the difference in MI between controls and ablated larvae is significant and we have corrected this. We reviewed all the figures again and found one other error which we corrected (S2d, not significant as per the text).

(4) Why use the terminology "epithalamic commissure" – is this different from habenular commissure?

We were trying to avoid confusion between abbreviations for caudal hypothalamus (Hc) and habenula commissure. In the revised version we eliminated abbreviations for the commissure, and changed the description to the "habenular commissure" throughout.

(5) Do the habenular ablations leave the PT commissural projections intact?

Yes: after unilateral habenula ablation images, confocal imaging demonstrates that *y279* fibers still run through the commissure. We added images to Figure S3h and added the text, "After unilateral

habenula ablation, *y279-Gal4* labeled fibers within the habenula commissure remained intact, consistent with our finding that these fibers originate from the PT (Figure S3h).”.

(6) Line 301: “In contrast, much less is known about how stochastic individual patterns of left/right identity are established”. True, but there are some good examples of this in worm and indeed in the fish habenulae, if the overriding influence of left-sided Nodal signalling is removed, it is proposed that stochastic differences in Fgf signalling can determine handedness.

Certainly, in the absence of lateralized Nodal expression, parapineal position randomizes to left or right in about 2/3 of embryos (see Roussigne et al, 2018). But the Nodal pathway here is part of a molecular mechanism that normally establishes a consistent left anatomical bias - we have added this point and reference to the Discussion. The most intensively studied paradigm for stochastic L/R asymmetry in the nervous system is the pair of AWC neurons in *C. elegans*. Here there is a solid understanding of the molecular signaling pathway downstream of the symmetry breaking event, and an understanding that Ca²⁺ signaling is involved in the event itself. But the symmetry breaking event remains mysterious. We have added a recent reference to the Discussion which discusses the AWC situation in detail.

Reviewer #2 (Remarks to the Author):

In this paper, Horstick and colleagues report the interesting observation that larval zebrafish display lateralized motor behavior upon sudden loss of illumination. They use this observation to identify neurons that influence lateralized motor behavior. Specifically, by lesioning random subsets of neurons using GAL4 lines, they suggest that neurons in the posterior tuberculum and habenula are part of a circuit influencing lateralized behavior. This is complemented by calcium imaging and laser ablation experiments. The authors additionally use genetics to identify a signaling pathway that influences motor asymmetry, although it is unclear how this is linked to the structures defined earlier.

We appreciate all the reviewer's ideas on strengthening the manuscript and have included them throughout.

There are a number of points that should be addressed to increase the impact of the paper.

1. The cells in the posterior tuberculum respond to light. Is there a direct retinal innervation to the posterior tuberculum? It would be interesting to inject a lipophilic tracer into the retina to see if there is direct connectivity with the cells in $\gamma 279$ or $\gamma 375$. How does the projection fit with previous reports e.g. Robles et al1 and Easter and Burrill2. In other words, what AF would this be?

Thank you, this is a good idea. We imaged $\gamma 279$ -Gal4;UAS:Kaede crossed with *atoh7:EGFP*, allowing visualization of rostral PT and RGC arborizations, respectively (after photoconversion of Kaede to red). The confocal images were added as Figure S2g-h, and the results added to the text "Since PT neurons were acutely responsive to changes in illumination, we asked whether PT neurons might receive photic information directly from the eye. We visualized retinal ganglion cell (RGC) termination zones (arborization fields, AF) in *atoh7:GFP* larvae, and found that rostral PT neurons were proximal to region AF3 (Figure S2g){Burrill (1994)}. RGC axons did not directly ramify within the rostral PT, however neurites from RGCs and the PT were closely apposed (Figure S2h)". To the following sentence we added "...and potentially receive direct photic input from the retina."

2. It should be noted somewhere in the discussion whether there are known homologies between the posterior tuberculum and brain regions in other vertebrates such as mammals. This would help place this work in a broader context. Information on connectivity (e.g. input from the retina) would help in this respect.

The posterior tuberculum is one of the most challenging parts of the fish brain whose embryonic origins are obscured by substantial migration within the diencephalon. Accordingly we are reluctant to make very strong claims about homology. However, we have added text to the results, pointing out that the PT is "the fish derivative of the basal plate of prosomere 3 (Figure 3e; see S2b-e for comparative neuroanatomical discussion)".

3. How does this work fit in with the response to loom described by Heap et al3? This should be considered in the discussion, especially with regards to lines 272-274. Do the PT neurons here project to the tectum?

This is a nice observation: although these PT neurons don't project to the tectum, there is an interesting similarity in function, and we have added text to the Discussion pointing this out "Intriguingly, the thalamus, dorsally adjacent to the PT, also relays directional information, sending information about visual dimming cues to the tectum (Heap et al 2018)".

4. Where possible, effect sizes should be indicated. In comparing behavior of groups, plotting differences would

be an effective way of presenting the data. Individual data points should be shown, e.g. in a jitter plot, where possible. (e.g. Fig. 1e)

We have added effect sizes throughout to the figure legends (Cohen's d, rank-biserial correlations, and partial ETA squared as appropriate). Individual data points are added for majority of figures, however jittered data points are excluded from match index figures as the entire data set is represented by 4 numbers (0, 25, 50, 75, 100) limiting the value of jittered points.

5. The commissure running between left and right habenula is the habenular commissure, not the epithalamic commissure.

We changed epithalamic to habenular throughout.

6. What is the significance of the vGlut label in Fig. 3f? Are y279 cells neither glutamatergic nor GABAergic (Fig. S2)? Before excluding the possibility that the cells are GABAergic, direct imaging of a double transgenic (eg y279 driving GFP and *gad1b:DsRed*) should be carried out.

As requested, as imaged y279-Gal4;UAS:Kaede, *gad1b:dsRed* triple larvae and have added a panel to Figure 3F comparable to the current vglut/y279 image. We added the sentence starting , “We did not observe co-expression of either an excitatory (*vglut2a*) or inhibitory neurotransmitter (*gad1b*) transgenic marker in rostral PT neurons (Figure 3f).”.

The following minor errors should be corrected:

1. Line 317: the phrase that mindbomb is “a ubiquitin ligase required for notch signaling” is already stated (line 315-316).

We deleted the redundant “ubiquitin....” from the text.

2. Line 481: the word “line” should be “objective” or “lens”

Sorry, this should have read 'line averaging' to describe the acquisition parameters. We have corrected this.

3. Line 484: “Fb” is undefined.

Fb was changed to F0 which is defined.

4. Line 771: In figure 6c, homozygous is on the left and heterozygous is on the right.

We fixed the labels in Figure 6c.

References

1. Robles, E., Laurell, E. & Baier, H. The retinal projectome reveals brain-area-specific visual representations generated by ganglion cell diversity. *Curr Biol* 24, 2085–2096 (2014).
2. Burrill, J. D. & Easter, S. S. Development of the retinofugal projections in the embryonic and larval zebrafish (*Brachydanio rerio*). *J Comp Neurol* 346, 583–600 (1994).
3. Heap, L. A. L., Vanwalleghe, G., Thompson, A. W., Favre-Bulle, I. A. & Scott, E. K. Luminance Changes Drive Directional Startle through a Thalamic Pathway. *Neuron* 99, 293–301.e4 (2018).

Reviewer #3 (Remarks to the Author):

In the manuscript, the authors described three types of motor asymmetries of larval zebrafish, including dark-induced circular swimming, two-alternative phototaxis, and acoustic startle behavior. They found that, for each larva, the left/right direction preference was consistent across the three types of behavior, sustained across contiguous trials and across several days, but was overridden by visual inputs. They also addressed the ontogeny of the asymmetry, which was neither dependent on the earlier visual experience nor inherited. Taken together, the authors characterized a robust motor asymmetry. Furthermore, the authors tried to examine the circuitry and molecular relevance of such asymmetry, including the PT-habenula pathway and mutations like *moe* and *mib*, but the results were not clear. Taken together, this is an interesting and somehow preliminary work on motor asymmetry and can be improved by well-designed mechanism studies.

We appreciate the reviewer's clear ideas on experiments to improve the paper and have carried them out.

Major concerns:

1. As the authors have shown in Fig. 1g, the left-preferring and right-preferring larvae were found with similar proportions. The authors may want to examine the individual difference between the left-preferring and the right-preferring larvae, to address the underlying mechanisms either on circuit or on molecular signaling, instead of comparing the preference distribution between the normal and the genetically modified population. It is important to classify the larvae according to their performance in the circular swimming and then examine their tubercle and habenula dark responses and their PT-habenula connection.

We performed PT calcium imaging experiments between left and right identified individuals and calcium imaging in $\gamma 279$ specified habenula neurons. The results are added to Figure 4; Figure S3j - see also Reviewer 1 comment 3 for further description). We do not see a difference in calcium in left/right PT neurons in fish classified as left/right motor biased.

We also performed photoconversion experiments to trace unilateral PT projections and measure the extent of arborization in the habenula, according to motor-bias and PT origin (Figure S2e-f). However, no differences emerged.

2. It had better provide information on how many Gal4 lines were screened and what brain areas were covered in these lines. Are there any other lines labeling anterior PT or affecting the motor asymmetry? Especially, as ablation of bilateral habenulae abolished the asymmetry, why such defect was not found on some Gal4 lines labeling the habenulae?

We added a supplementary table listing the Gal4 lines ablated and major neuroanatomical structures labeled by these lines. Only $\gamma 279$ and $\gamma 375$ labeled the rostral PT. The habenula comprises at least 18 neuronal types (Pandy Curr Biol (2018)), so although 3 other lines labeled habenula neurons it is not very surprising that ablation failed to disrupt motor bias.

3. In Fig. 3b, the authors showed that the difference in Match Index between $\gamma 375$ and control was not significant. Thus they should provide rationale on describing 'a decrease in motor bias after ablation'?

We apologize, the Figure was mislabeled and the text is correct: the MI difference between $\gamma 375$ ablated/control is significant.

4. The authors found that motor bias was decreased in two Gal4 lines, $\gamma 279$ and $\gamma 375$. The authors then

postulated that neurons affected in both lines should be responsible for the motor bias. This may have missed other candidates, as the non-overlapping neurons in the two lines may reside in the same pathway innervating the motor bias effector, or they may work synergistically in other manner.

We were agnostic as to whether an overlapping population would exist or that if it did exist, it would label the key neurons, and only wrote "We reasoned that clusters labeled by both lines would be strong candidates for driving lateralized behavior." The laser ablation experiment confirmed that the overlapping population are important for motor bias, but the reviewer is correct, either line may label additional synergistic neurons (in fact in $\gamma 279$ the habenula neurons are precisely such a population).

5. Although $\gamma 279$ ablation reduced the motor asymmetry, it is possible that anterior PT neurons may just respond to the dark signal unilaterally and initiate the turning motor behavior. This can explain both that these neurons are necessary and that the no left/right difference was found in their activities. Besides, it would be better if the authors can show acoustic startle assay results or two-target phototaxis assay results of these larvae.

This is a good idea, but critically, after $\gamma 279$ or $\gamma 375$ ablation of PT neurons, the total amount of turning behavior was not reduced (Fig. S2a), so these are not the neurons that drive turning behavior, they just impose direction.

As requested, we performed the startle assay on unilaterally PT ablated larvae and found that motor bias is lateralized in the same way as circling behavior (new Fig. 3i).

6. Alternatively, the PT neurons may compete or their downstream neurons, like in the habenulae, may compete in driving asymmetric behavior. It may be quite informative to examine such competition and compare between the left- and the right- preferring larvae, like figuring out the PT-habenula connection. Besides, it would be better if the authors can free the tail or record electromyography (EMG) or do calcium imaging on motor neuronal activities on the restrained larvae, to examine the neurons contributing to the generation of motor asymmetry.

We agree that identifying downstream targets of the PT neurons will be informative, but the proposed experiments are not feasible because as we noted in the results, we do not observe sustained directional tail movement in embedded tail excised larvae, precluding EMG recordings or calcium imaging on motor populations. Because our ablation data supports a role for PT neurons in motor bias in free-swimming larvae, we also performed an experiment using pERK staining after circling behavior. However, again, we did not see an activity difference in left/right biased larvae, almost certainly because the low temporal resolution of this method leads to the signal being swamped by other behaviors occurring in the same time window.

7. The authors found weak motor bias on the $\gamma 606$ mutant. While this line may be quite informative in examining the asymmetry mechanism beyond the morphological abnormality, the authors went on to examine its relationship to the Nodal pathway. As the Nodal pathway affects left-right asymmetry development, which was not found in previous left- or right- preferring larvae, such experiments lack sufficient rationale.

Heterozygous mutations in this pathway have not previously been shown to affect left/right asymmetry development, and certainly not motor bias. We think this is an exciting result with significant implications for researchers working on the genetic basis of handedness and other motor asymmetries. However, in deference to the reviewer, we have minimized text throughout the manuscript concerning the molecular characterization of Notch haploinsufficiency on motor bias including removing the discussion concerning Notch regulation of Nodal.

Minor concerns:

1. What's the ecological significance of the motor asymmetry found in dark but not in light?

In the discussion paragraph starting with, "Even less clear is the nature of the ethological...". we outline our thoughts of why the motor asymmetry may exist and why different patterns of locomotor activity are observed during illuminated and dark conditions. Briefly, we think that efficient motor selection requires functional neural asymmetry. In the light this is supplied by environmental cues. In the dark, intrinsic bias provides a lateralized drive to overcome difficulty in selecting a response direction.

2. In determining the visual contribution, the authors employed *atoh7* mutants. While RGB inputs were abolished in this line, there is also possible that the deficiency in other brain areas expressing *atoh7*, like hindbrain nuclei, played their roles.

Thank you, this is a good point. We have performed a dual enucleation experiment, and tested motor bias. Confirming the *atoh7* mutant result, enucleated larvae also exhibited individual motor bias after loss of light. We added Figure S1j with this result and in the results, added that we tested both *atoh7* mutants and enucleated larvae

3. It deserves examination how the visual information predominates over intrinsic motor asymmetry.

Unlike humans which show a very strong motor asymmetry, motor asymmetry in larvae is only a bias. The Match Index in most experiments indicated that larvae have only a 70% likelihood of pursuing their 'preferred' direction. It does not seem unreasonable that visual cues could easily over-ride such a bias.

4. For larvae tested at 7 or 10 dpf, there were some larvae which changed their direction preference (Fig. 1j). How such did changes happen?

Because our finding of motor bias in larval zebrafish larvae is novel and to many, unexpected, we wanted to use a simple and conservative criterion for classifying motor bias. Therefore we simply classified each larvae as left or right-biased based on the first trial performed by that larva. However, as we show in Figure 1E, for any given larva, 30% of events are performed in the opposite direction of the identified bias. This intrinsic variation results in miscategorization of 30% of individuals on trial 1. Larvae which apparently changed direction preference were likely mis-classified due to stochastically swimming in the non-preferred direction on trial 1.

5. In the reference session, the journal names were in abbreviation, like 67, or in full , like 68. The authors may use a consistent style.

Fixed.

Reviewers' comments:

Reviewer #1 (Remarks to the Author):

The authors have done a first-rate job of addressing all our concerns with additional experiments and analyses as well as refining the manuscript. We feel this is an important piece of work for the zebrafish and the wider neuroscience community, as it provides unique insights regarding the neural pathway and potential genetic mechanisms underlying behavioural asymmetry. We only have one minor comment to add:

Figure 7: The subpanels (perhaps after labelling them a,b,c and d) highlighting differences between intact PT-Hb pathway, unilateral/bilateral ablations and enucleation should be explained in the Discussion (Pages 14-15); it is a nice explanation of how motor asymmetry could emerge, for example, and how visual information could override this intrinsic bias.

We are happy to proceed with accepting the manuscript in its current form. Congratulations to the authors on a very interesting paper!

Reviewer #2 (Remarks to the Author):

The authors have addressed all my comments in the revision.

Reviewer #3 (Remarks to the Author):

We appreciate all the works the authors have done in addressing the possible our concerns and agree that the current manuscript is worth of publishing on Nature Communications. Although no clear conclusion could be drawn on the circuitry mechanism underlying motor asymmetry, this manuscript has provided sufficient information on what can be obtained with the popular techniques, especially given the difficulty of replicating the behavior on restrained fish. However, we would like to suggest that more helpful information can be extracted, according to how PT neurons encode motor-bias information beyond simply position determination, as we raised in previous point 6. As PT is a large area consisting of 18 types of neurons, there should be more information to be mined beyond dividing PT neuron as left and right and averaging within in each hemisphere. Even for mapping the response spatial distribution, coding the response amplitude of each neuron by color would be more informative than what is shown in Fig. 4b of the current version. Furthermore, the authors may also want to consider the response dynamics. They may also perform some immunohistochemical experiments to characterize the neurochemical components of the PT-Hb connection and shed light on the possibly specific circuit contributing the motor asymmetry.

Reviewer #1 (Remarks to the Author):

The authors have done a first-rate job of addressing all our concerns with additional experiments and analyses as well as refining the manuscript. We feel this is an important piece of work for the zebrafish and the wider neuroscience community, as it provides unique insights regarding the neural pathway and potential genetic mechanisms underlying behavioural asymmetry. We only have one minor comment to add: Figure 7: The subpanels (perhaps after labelling them a,b,c and d) highlighting differences between intact PT-Hb pathway, unilateral/bilateral ablations and enucleation should be explained in the Discussion (Pages 14-15); it is a nice explanation of how motor asymmetry could emerge, for example, and how visual information could override this intrinsic bias.

We are happy to proceed with accepting the manuscript in its current form. Congratulations to the authors on a very interesting paper!

We appreciate the reviewer's support of our work. We have labeled Figure 7 as suggested and included additional explanations of the model in the discussion.

Reviewer #2 (Remarks to the Author):

The authors have addressed all my comments in the revision.

We again thank the reviewer for thoughtful criticism and suggestions.

Reviewer #3 (Remarks to the Author):

We appreciate all the works the authors have done in addressing the possible our concerns and agree that the current manuscript is worth of publishing on Nature Communications. Although no clear conclusion could be drawn on the circuitry mechanism underlying motor asymmetry, this manuscript has provided sufficient information on what can be obtained with the popular techniques, especially given the difficulty of replicating the behavior on restrained fish. However, we would like to suggest that more helpful information can be extracted, according to how PT neurons encode motor-bias information beyond simply position determination, as we raised in previous point 6. As PT is a large area consisting of 18 types of neurons, there should be more information to be mined beyond dividing PT neuron as left and right and averaging within in each hemisphere. Even for mapping the response spatial distribution, coding the response amplitude of each neuron by color would be more informative than what is shown in Fig. 4b of the current version. Furthermore, the authors may also want to consider the response dynamics. They may also perform some immunohistochemical experiments to characterize the neurochemical components of the PT-Hb connection and shed light on the possibly specific circuit contributing the motor asymmetry.

We thank the reviewer for recognizing our efforts to address their previous comments. At the reviewer's request we have further mined the PT calcium imaging data and added new analysis looking at response dynamics between left and right biased individuals - but do not see significant differences. We added the data into Figure S2, and the text is updated to reflex the new analysis. Note that 4b already has the response amplitude of each neuron coded by color. Regarding activity in PT neurons, the basic problem is that the behavior is not manifest in head-embedded fish. Our primary motivation for activity analysis was to determine whether these neurons are active after light OFF. That we do not see biased activity in PT neurons may be because embedding eliminates biased activity in PT neurons, or it could be because on any specific trial, there is a 30% chance of producing a response in the non-preferred direction, and without being able to monitor the behavior, we can't identify those 'contaminating' trials.

Finally, the reviewer is correct in stating that the PT is a large neuroanatomical structure. However, the bias maintaining neurons are a small cluster of 60 neurons at the very rostral-most portion of the PT - we have added text to the Results to clarify this point. Each of the bilateral clusters is within a 30 micron sided-cube ; for context this volume is only about 20% that of the right habenula. Ascertaining the neurochemical identity of rostral PT neurons would be informative: in Figure 3, we show these cells are not glutamatergic or GABAergic, and in Figure S2 show these neurons do not overlap with 3 neuropeptides known to expressed in the vicinity, or the well-known TH+ dopaminergic population. Our future efforts at characterizing these neurons will rely on sequencing approaches, but this is beyond what we can do here.

REVIEWERS' COMMENTS:

Reviewer #3 (Remarks to the Author):

All my comments have been well addressed. I have no more comment and think it is ready for publication.